# Harnessing multivalency and FcγRIIB engagement to augment anti-CD27 immunotherapy

Marcus A. Widdess, Anastasia Pakidi, Hannah J. Metcalfe, H. T. Claude Chan, Tatyana Inzhelevskaya, Chris A. Penfold, C. Ian Mockridge, Steven G. Booth, Sonya James ⑩, Sean H. Lim ⑩, Stephen A. Beers ⑩, Mark S. Cragg ⑩ & Aymen Al-Shamkhani ⑩ ✉

Despite significant clinical progress, checkpoint blockade remains limited by variable response rates, resistance, and toxicity. Activating costimulatory receptors offers a promising alternative to enhance anti-tumor immunity. However, there is insufficient understanding of how to mimic physiological membrane-anchored costimulatory ligands. Here, we describe a strategy for developing effective agonists of the costimulatory receptor CD27 by increasing both antibody valency and FcγRIIB engagement. Engineered anti-CD27 antibodies capable of tetravalent binding to CD27 and selective FcγRIIB association exhibit potent T cell stimulatory activity and anti-tumor efficacy in pre-clinical models, compared to bivalent counterparts. The anti-tumor effects of the tetravalent antibody are mediated through CD8[+] T cell activation without evidence of regulatory T cell depletion. Mechanistically, whereas the increase in avidity drives more efficient CD27 clustering, FcγRIIB engagement triggers polarization of receptor clusters to the cell-cell interface and reduces receptor internalization. This work provides a framework for developing more effective agonist-based T cell stimulatory therapies.

T cells express unique, somatically rearranged T cell receptors (TCRs) that exhibit a degree of cross-reactivity, thus enabling the recognition of diverse pathogen and tumor-derived antigens. To maintain immunological homeostasis, additional receptor-ligand interactions regulate T cell activation through costimulatory or inhibitory signals[1,2]. It has now been established that immune evasion is a key mechanism by which tumors grow and metastasize[3]. Immune suppression within the tumor microenvironment skews the balance of inhibitory and costimulatory signals towards inhibition, leading to T cell ignorance, anergy, and exhaustion[4]. Antibody-mediated blockade of inhibitory immune checkpoints has shown clinical efficacy across multiple cancer types, with CTLA-4 and PD-1 inhibitors demonstrating particular success[5]. However, challenges persist, including variable response rates, development of resistance, and immune-related adverse events[5].

An alternative approach to enhance anti-tumor immunity involves the direct activation of costimulatory receptors. The premise underpinning this strategy stems from the original work demonstrating that activation of CD28 through ectopic B7 expression on tumor cells leads to their elimination by the immune system[4,6]. Similar findings with other costimulatory ligands and agonist antibodies targeting TNF receptor superfamily (TNFRSF) members have supported the development of clinical agents[4,7]. However, the majority of clinical agonists have demonstrated insufficient therapeutic activity, the precise reasons for which remain poorly understood[7]. Multiple factors have been shown to influence the activity of agonist anti-TNFRSF antibodies, including epitope, isotype, and affinity[8–10]. Receptor clustering by antibodies is generally required for activation of downstream signaling events, a process that mimics the natural interaction of TNFRSF

Antibody and Vaccine Group, Centre for Cancer Immunology, School of Cancer Sciences, Faculty of Medicine, University of Southampton, Southampton, UK.
✉e-mail: aymen@soton.ac.uk

proteins with their trimeric ligands. Thus, murine IgG1 antibodies targeting members of the TNFRSF exert strong agonism dependent on binding to the inhibitory Fcγ receptor (FcγRII) and introduction of mutations into human IgG1 Fc that enhance affinity towards human FcγRIIB similarly bolster agonism[8,11–13]. Moreover, various antibody engineering approaches have been proposed that enhance TNFRSF agonism by increasing antibody avidity[14–16], or conformational stability[17,18].

Antibodies that target the T cell costimulatory receptor CD27 have shown promise in enhancing anti-tumor immunity in preclinical models[19–22]. The efficacy and mechanism of action have been shown to be influenced by antibody isotype. For example, the activity of an agonistic anti-CD27 antibody, MK-5890, in both primary human tumor cultures and human CD27 knock-in mice, was significantly diminished when the isotype was switched from human IgG1 to IgG2 or a variant that further reduces the interaction with FcγR[23]. However, targeting CD27 with human IgG1, an isotype associated with strong effector function, resulted in depletion of T cells, especially regulatory T cells, calling into question the choice of using the human IgG1 backbone in CD27 agonists[23–25]. Given these uncertainties and the limited clinical efficacy of Varlilumab, a human IgG1 anti-CD27 antibody[25,26], we sought strategies to generate more effective antibodies with enhanced therapeutic potential.

Here, we explore approaches to optimize the agonistic activity of CD27-targeted antibodies and investigate how valency and FcγR engagement shape their activity in both mouse and human systems. Our data show that these mechanisms act synergistically to enhance agonism and anti-tumor efficacy. Together, these findings outline a strategy for developing therapeutic antibodies that achieve optimal activation of anti-tumor T cells.

## Results
### Characterization and in vitro activity of a tetravalent anti-mouse CD27 antibody

To assess the impact of increased valency on anti-CD27 antibody potency, we engineered a tetravalent version of the anti-mouse (m) CD27 mAb AT124-1 by incorporating additional Fab arms into its structure (Fig. 1A). Antibodies were produced as mouse IgG1, as this isotype is known to potentiate agonism by anti-TNFRSF antibodies in mice through FcγRII-mediated crosslinking[24,27,28]. To investigate the requirement for FcγR engagement for bivalent and tetravalent reagents, we also produced anti-mCD27 antibodies incorporating the N297Q (NQ) mutation, which abrogates FcγR binding[29]. Size-exclusion chromatography (SEC) confirmed that all antibodies existed as monomers in solution, free from aggregates (Supplementary Fig. 1A), and SDS-PAGE analysis established that heavy and light chains of the anticipated size were produced (Supplementary Fig. 1B). The structure of tetravalent anti-mCD27 was also characterized by negative-staining electron microscopy. Class average image analysis revealed the tetrameric structure of the antibody, displaying four distinct Fab sections (Fig. 1B). The outer Fab sections exhibited a degree of flexibility, consistent with the predicted mobility of the peptide linker connecting them to the inner Fab section (Fig. 1B). Next, we employed surface plasmon resonance (SPR) to investigate the effect of increased valency on the binding strength to mCD27. Our results demonstrated a >5-fold increase in apparent affinity, primarily attributed to a slower dissociation rate of the tetravalent antibody from the receptor (Fig. 1C and Table 1). We also investigated the effects of increased valency on the binding of anti-mCD27 antibodies to mCD27 expressed on the surface of Jurkat cells and observed a significant increase in avidity, as reflected by a ~100-fold decrease in the EC50 value for the tetravalent format (Fig. 1D). Next, we assessed if increased valency and avidity translated into enhanced NF-κB activation, a key downstream signaling pathway of CD27[30,31]. When applied in solution, neither bivalent nor tetravalent anti-mCD27 antibodies induced GFP expression in mCD27⁺

Jurkat NF-κB GFP reporter cells (Fig. 2A, left panel). However, both formats induced NF-κB signaling when mCD27⁺ reporter cells were co-cultured with Chinese hamster ovary (CHO) cells expressing mouse FcγRII (Fig. 2A, right panel and 2B). Importantly, whereas at high concentrations (10–100 nM) the two formats stimulated similar levels of NF-κB activation, tetravalent anti-mCD27 was by far the most active at concentrations ranging from 0.001 to 1 nM (Fig. 2A, right panel). The enhanced potency of tetravalent anti-mCD27 when compared to bivalent anti-mCD27 and the requirement for Fc-mediated crosslinking was similarly observed when OT-I T cell proliferation under conditions of sub-optimal peptide stimulation was used as the readout for CD27 signaling (Fig. 2C). Collectively, our in vitro data suggest that the combination of tetravalency and FcγRII binding produces a level of activation not achievable with bivalent antibodies.

### Impact of valency and FcγRII binding on in vivo antigen-specific CD8⁺ T cell responses and immunotherapy with anti-mCD27 antibodies

To explore the broader implications of our in vitro data, we initially compared the ability of bivalent and tetravalent anti-mCD27 antibodies to boost a CD8⁺ T cell response in vivo. This was first addressed using adoptively transferred CD45.1⁺ OT-I T cells and administration of ovalbumin 257–264 peptide (OVA$_{257-264}$) combined with either bivalent or tetravalent anti-mCD27 antibodies. Figure 3A–C demonstrates that tetravalent anti-mCD27 treatment led to significantly greater CD8⁺ T cell expansion compared to bivalent anti-mCD27 or an isotype control. This enhanced T cell proliferation was sustained throughout the contraction phase. Moreover, the "Fc silent" N297A variants of the bivalent and tetravalent anti-mCD27 antibodies did not induce significant CD45.1⁺ CD8⁺ T cell expansion, indicating that FcγR crosslinking was required for activity of anti-CD27 antibodies in vivo, thus corroborating our in vitro findings. Of note, this enhanced in vivo activity of the tetravalent anti-CD27 mIgG1 could not be attributed to improved pharmacokinetics, as it exhibited a slightly shorter plasma half-life compared to its bivalent counterpart (Supplementary Fig. 2).

Next, we compared the activity of anti-mCD27 antibodies in an immunotherapy setting wherein mice bearing a B16 melanoma variant that expresses OVA as a surrogate tumor antigen (B16-OVA) were treated with OVA and the antibodies indicated in Fig. 4. Mice were then assessed for endogenous OVA-specific CD8⁺ T cell expansion and tumor growth. Consistent with the OT-I T cell responses, tetravalent anti-mCD27 elicited the strongest endogenous OVA-specific CD8⁺ T cell response, surpassing that of the bivalent format, and this response was sustained throughout the sampling period (Fig. 4A–C). This enhanced T cell response correlated with more effective tumor control by tetravalent anti-mCD27 (Fig. 4D). In contrast, but in keeping with their muted OVA-specific CD8⁺ T cell responses, the Fc-inert mAb did not elicit robust tumor control. These experiments confirm that the tetravalent anti-mCD27 antibody induced a stronger endogenous OVA-specific CD8⁺ T cell response and improved tumor control; although, due to the aggressiveness of B16-OVA tumors and the likely emergence of OVA-loss variants[32], tumor growth was ultimately not fully prevented (Fig. 4D and Supplementary Fig. 3).

Next, we conducted experiments to compare bivalent and tetravalent anti-mCD27 antibodies in a second model, the BCL$_1$ lymphoma, which lacks an identified tumor-specific antigen, and in which CD27 agonism had previously been shown to elicit protective T cell responses[19,22,24,33]. As the Fc-inert anti-CD27 antibodies showed limited activity, we focused on testing antibodies with functional mIgG1 Fc in this model. Bivalent anti-mCD27 provided modest protection from the BCL$_1$ tumor, increasing median survival from 16 days in the isotype control group to 39.5 days (Fig. 5A), which was further increased with the tetravalent anti-mCD27 antibody, with median survival increasing to 71.5 days (Fig. 5A).

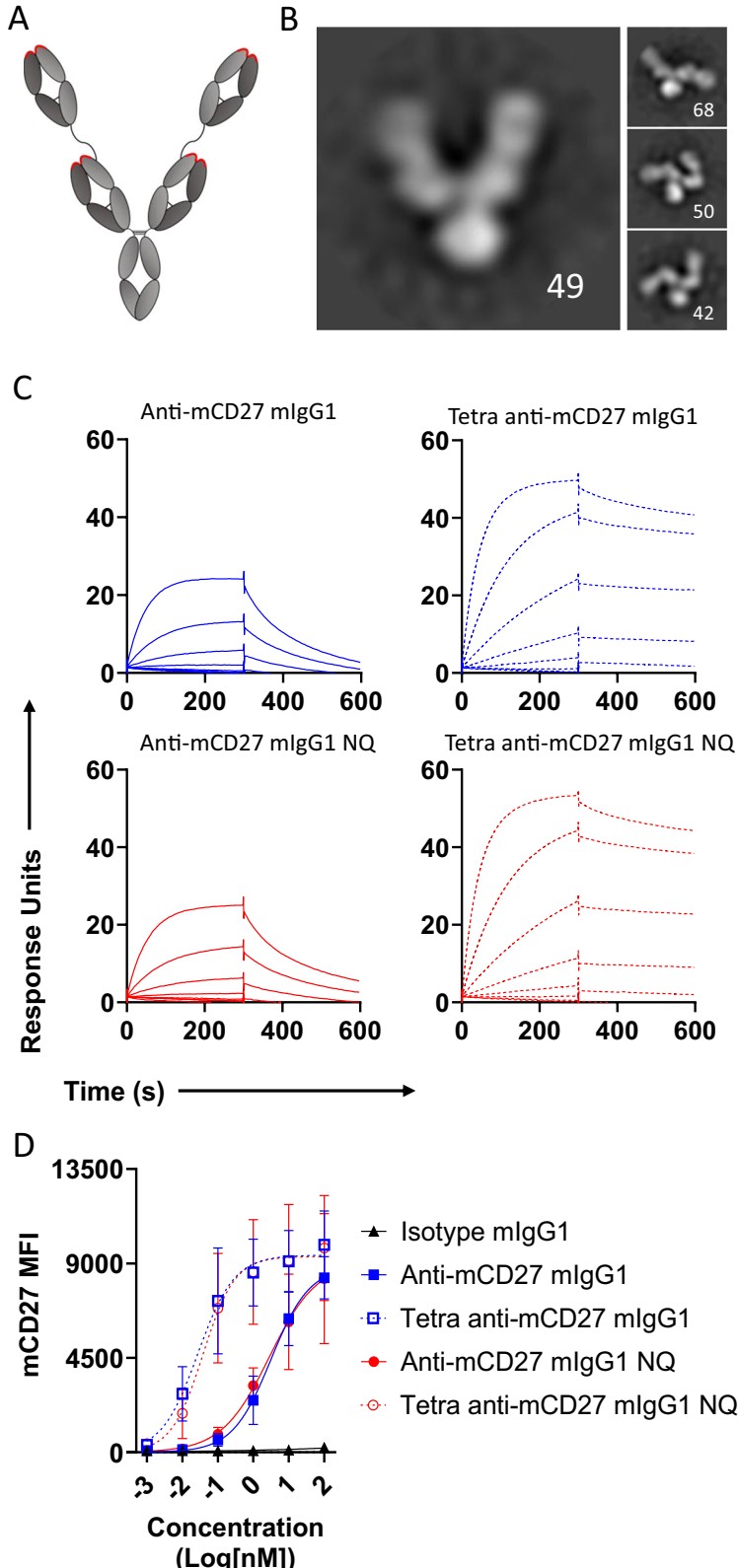

**Fig. 1 | Tetravalency enhances anti-mCD27 avidity for mCD27. A** Schematic representation of the tetravalent (Tetra) antibody, with CDRs indicated in red. **B** Representative class averages (n = 49, 68, 50, and 42 per image) from negative stain electron microscopy of Tetra anti-mCD27. **C** SPR analysis of anti-mCD27 antibodies. The indicated antibodies were injected over immobilized mCD27-Fc for 300 s at a starting concentration of 3.7 nM and then a 3-fold serial dilution thereof. Data show concentrations from one experiment. **D** mCD27+ Jurkat cells were stained with anti-mCD27 antibodies, with bound antibody detected using an APC-conjugated anti-mouse Fc secondary F(ab)$_2$. Secondary antibody binding was detected using flow cytometry. Data points are the mean ± SEM (n = 3) from 3 independent experiments. Source data are provided as a Source data file.

**Table 1 | Summary of the binding kinetics of anti-mCD27 antibodies**

| Antibody | $K_a$ ($\times 10^6\,M^{-1}\,s^{-1}$) | $K_d$ ($\times 10^{-3}\,s^{-1}$) | Apparent affinity ($K_D$, $\times 10^{-9}\,M$) |
|---|---|---|---|
| Anti-mCD27 mIgG1 | 1.3 | 12.7 | 9.5 |
| Anti-mCD27 mIgG1 NQ | 1.1 | 14.4 | 13.1 |
| Tetra anti-mCD27 mIgG1 | 1.6 | 2.9 | 1.8 |
| Tetra anti-mCD27 mIgG1 NQ | 1.6 | 2.4 | 1.5 |

Kinetic fits for anti-mCD27 antibody binding were calculated using the Biacore kinetics summary v3.1 software and the bivalent (2:1) binding model fit to the data in Fig. 1C. Source data are provided as a Source data file.

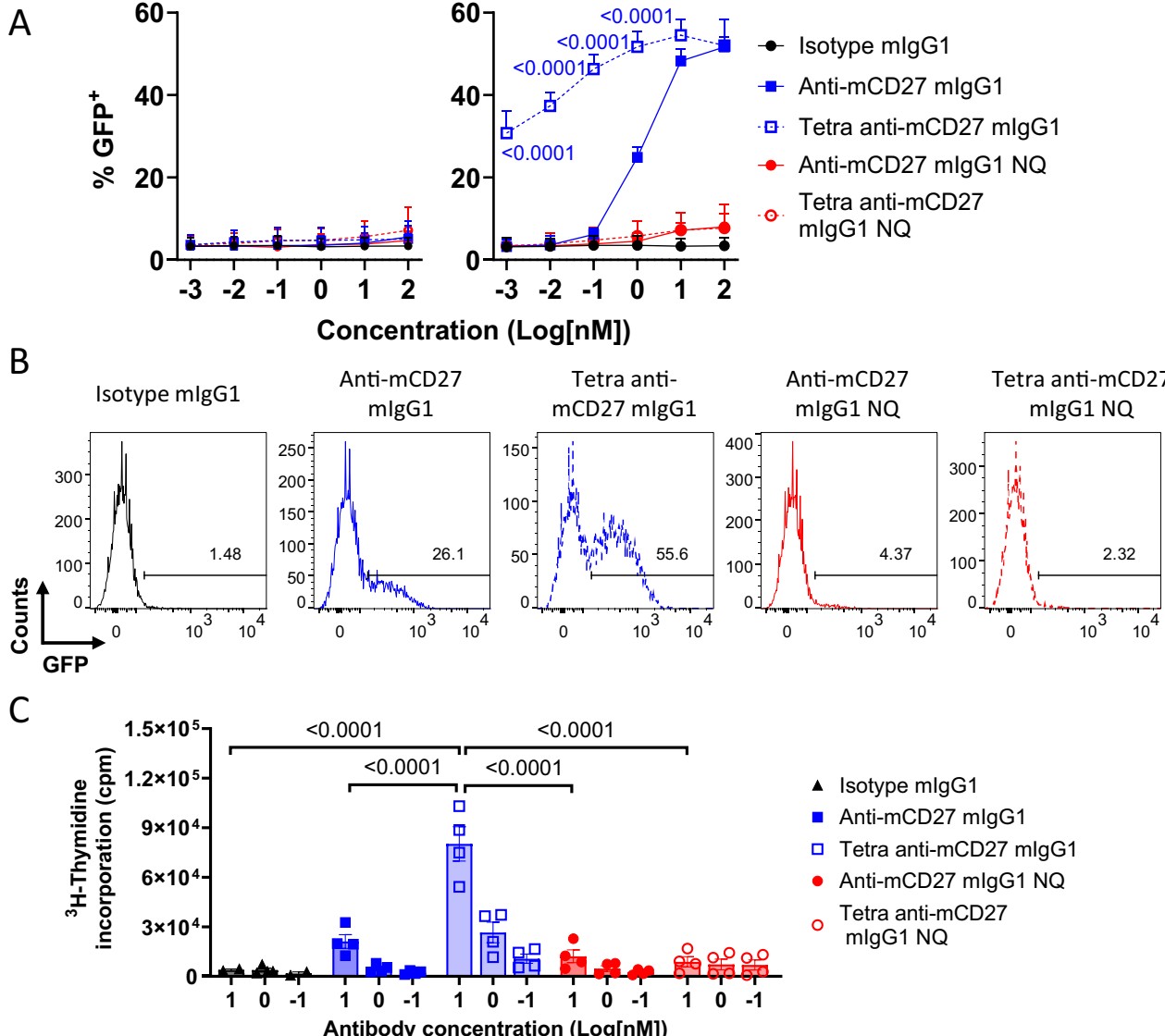

**Fig. 2 | Tetravalent anti-mCD27 induces potent agonism in the presence of mFcγRII. A**, **B** mCD27$^+$ Jurkat NF-κB GFP reporter cells were co-cultured with either wild-type (WT CHO) or mFcγRII$^+$ CHO cells in the presence of the indicated antibodies. NF-κB activation was quantitated by GFP expression using flow cytometry. **A** GFP production from Jurkat cells after co-culture with WT CHO cells (left panel) or mFcγRII$^+$ CHO cells (right panel) at the indicated antibody concentrations. Data points are the mean ± SEM (n = 3) from 3 independent experiments. Statistical significance was determined by one-way ANOVA (Tetra anti-mCD27 mIgG1 versus anti-mCD27 mIgG1), with significance values indicated in the figure. **B** Representative histograms of GFP production at 1 nM of the indicated antibodies when mCD27$^+$ Jurkat NF-κB GFP cells were co-cultured with mFcγRII$^+$ CHO cells.

**C** Naïve splenocytes were stimulated with OVA$_{257-264}$ peptide in the presence of the indicated antibodies for 72 h. T cell proliferation was assessed by measuring the incorporation of $^3$H-thymidine for the final 16 h of culture. Each data point is the mean of triplicate measurements. Bars are the mean ± SEM (n = 4) from 4 independent experiments, except Isotype control mIgG1 at 10 and 0.1 nM, which show mean ± SEM (n = 2) from 2 independent experiments. Statistical significance was determined by one-way ANOVA with Tukey's post-hoc test for multiple comparisons, and significance values are indicated in the figure. Responses of "NQ" antibody variants were not statistically significant compared to the isotype control. Source data are provided as a Source data file.

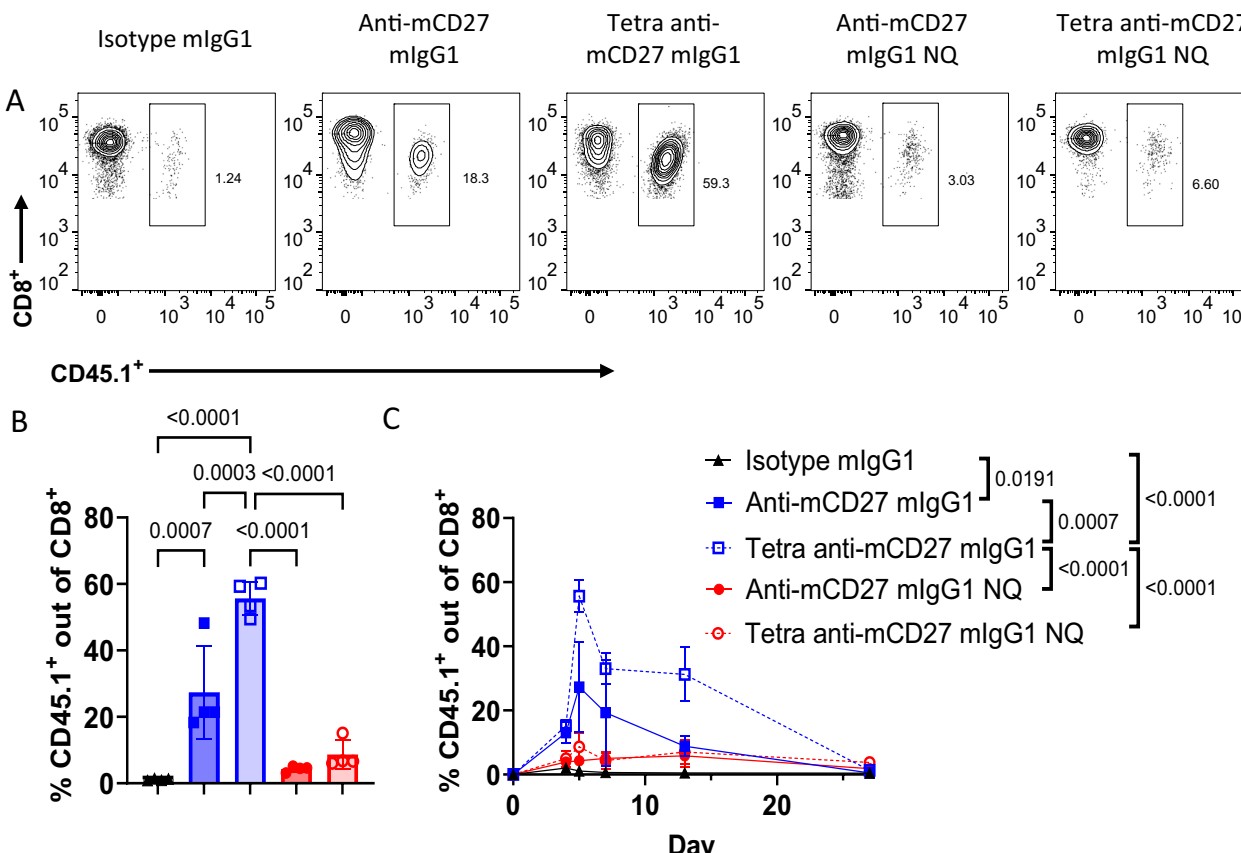

**Fig. 3 | Optimal priming of antigen-specific CD8⁺ T cells by anti-mCD27 requires tetravalency and FcγR engagement. A–C** C57BL/6 mice received OT-I Tg CD45.1⁺ cells on day 0, followed by OVA$_{257-264}$ peptide and the indicated antibodies on day 1. Expansion of OVA$_{257-264}$ specific CD45.1⁺ CD8⁺ T cells was monitored by flow cytometry of peripheral blood samples following staining with anti-CD8 and anti-CD45.1. **A** Representative flow cytometry contour plots obtained on day 5 and **B** the compiled data with each bar representing the mean ± SEM from 4 individual mice from one experiment (n = 4 mice per group). Statistical significance was determined by one-way ANOVA, with Tukey's post-hoc test for multiple comparisons, and significance values are indicated in the figure. **C** Time course of CD45.1⁺ CD8⁺ T cell expansion. Data points are the mean ± SEM from 4 individual mice from one experiment (n = 4 mice per group). Statistical significance was assessed by one-way ANOVA with Tukey's post-hoc test for multiple comparisons on areas under the curve (AUC), with significance values indicated in the figure. Responses of "NQ" antibody variants were not statistically significant compared to the isotype control. Source data are provided as a Source data file.

Given that BCL$_1$ lymphoma grows primarily in the spleen and FcγRII is abundantly expressed on both malignant and normal B cells[34], we next tested the tetravalent anti-mCD27 antibody in the CT26 colon carcinoma model, where FcγRII expression is absent from the tumor and limited to the tumor-infiltrating immune cells. CT26 tumors are known to exhibit heterogeneous responses to immunotherapy with anti-PD-1 antibodies, typically segregating into responders and non-responders[35]. We observed a similar pattern in mice bearing established CT26 tumors treated with anti-mCD27 antibodies. Strikingly, while the bivalent anti-mCD27 mIgG1 achieved a cure rate of only 50% among responders, the tetravalent anti-mCD27 antibody led to complete tumor regression in all responding mice, highlighting its superior therapeutic potential (Fig. 5B–D). Moreover, administration of a cumulative dose of 660 µg, equivalent to ~30 mg/kg of tetravalent anti-mCD27, did not induce weight loss in mice, suggesting that the enhanced anti-tumor activity was not associated with increased toxicity (Supplementary Fig. 4A). Furthermore, treatment with tetravalent anti-mCD27 mIgG1 did not increase alanine aminotransferase (ALT) or aspartate aminotransferase (AST) enzymatic activity in the blood, both of which are key indicators of liver toxicity (Supplementary Fig. 4B).

To understand the cell types involved in the anti-tumor response exerted by tetravalent anti-mCD27, we profiled T cell subsets within the tumor. Data presented in Supplementary Fig. 5 demonstrates that treatment with tetravalent anti-mCD27 induced an increase in both total and activated (4-1BB⁺ and Granzyme B⁺) CD8⁺ T cells without impacting total CD4⁺ or Foxp3⁺ CD4⁺ regulatory T cells. These data support the hypothesis that tetravalent anti-mCD27 exerts its effects through direct co-stimulation of CD8⁺ T cells.

## Tetravalency and FcγRIIB engagement synergize for optimal immune stimulation by anti-human CD27 antibody

To evaluate the clinical relevance of our findings, we explored whether increased valency and FcγRIIB interactions could potentiate the activity of anti-human (h) CD27 antibodies. To this end, we produced the agonistic anti-hCD27 antibody, hCD27.15[36], in a standard bivalent human IgG1 format and in a tetravalent configuration similar to the tetravalent anti-mCD27 antibody. Our previous work has shown that this clone recognizes an epitope distinct from that targeted by the clinical antibody Varlilumab and binds to human CD27 with moderate affinity, exhibiting stronger stimulatory activity[37]. Furthermore, to assess the impact of different FcγR in mediating antibody crosslinking, bivalent and tetravalent anti-hCD27 antibodies were generated using either a wild-type hIgG1 backbone, a hIgG1 variant (V11) with a higher FcγRIIB and a lower FcγRI affinity[12], or a variant (NA) that has compromised binding to FcγRs. SEC analysis confirmed that all anti-hCD27 antibodies were monomeric in solution and free from aggregates (Supplementary Fig. 6A). In addition, the overall structure of the tetravalent anti-hCD27 was verified by negative-staining electron

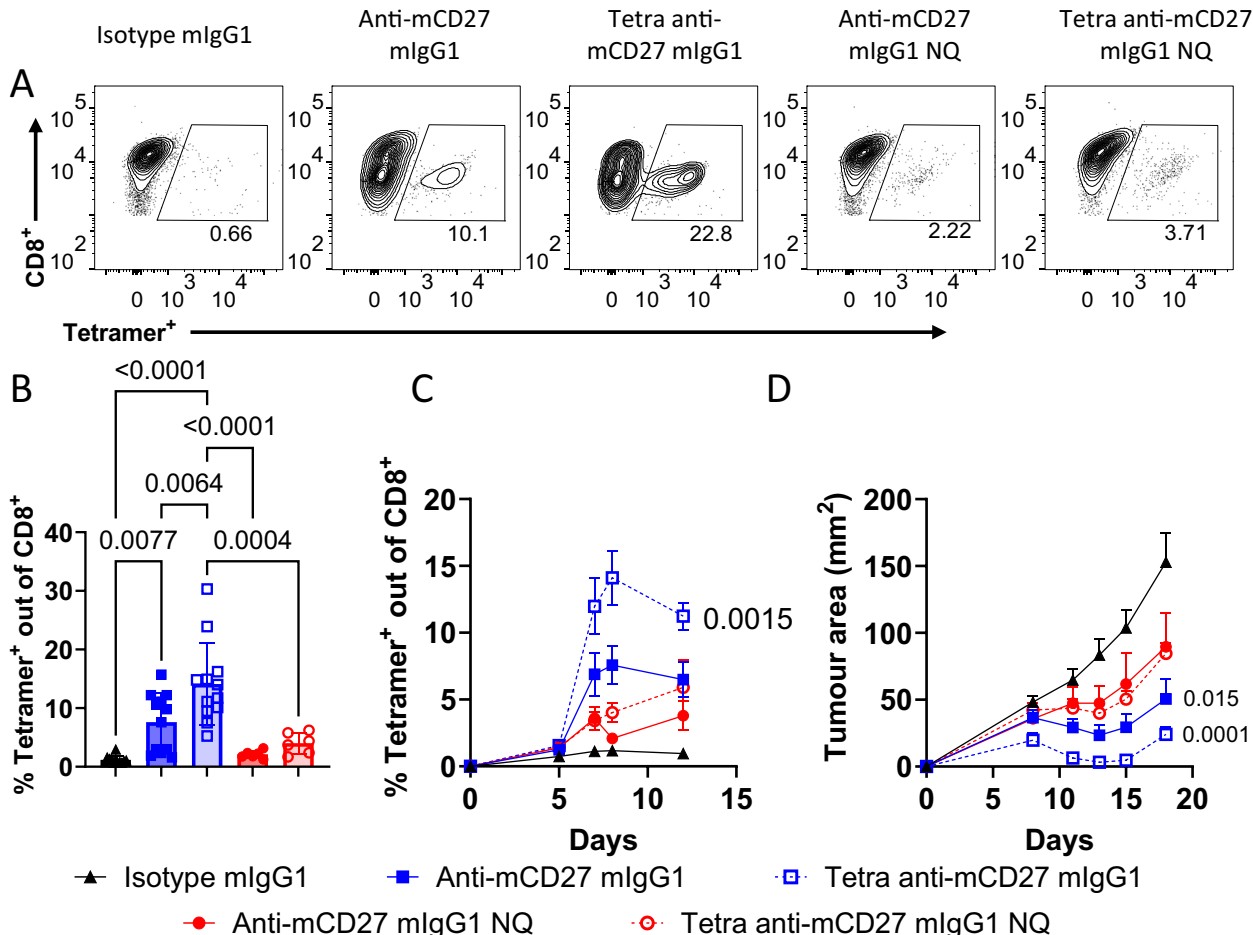

**Fig. 4 | Tetravalent anti-mCD27 generates superior vaccine responses compared to its bivalent counterpart. A–D** C57BL/6 mice were inoculated with s.c. B16-OVA melanoma on day 0 and treated with OVA and the indicated antibodies on day 1. Expansion of endogenous antigen-specific CD8⁺ T cells was monitored by obtaining peripheral blood samples from day 0 to day 12 and staining with anti-CD8 and H2-K$^b$ · OVA$_{257-264}$ tetramer (tetramer⁺). **A** Representative flow cytometry contour plots and **B** compiled data of the antigen-specific T cell response on day 7, with each bar representing the mean ± SEM from 12 individual mice (n = 12, isotype mIgG1, anti-mCD27 mIgG1, tetra anti-mCD27 mIgG1) across 2 independent experiments or 6 individual mice (n = 6, anti-mCD27 mIgG1 NQ, tetra anti-mCD27 mIgG1 NQ) from 1 experiment. Statistical significance was determined using one-way ANOVA, with Tukey's post-hoc test for multiple comparisons, and significance values are indicated in the figure. **C** Time course of the antigen-specific T cell

response, with each data point representing the mean ± SEM (n = 12, isotype mIgG1, anti-mCD27 mIgG1, tetra anti-mCD27 mIgG1; n = 6, anti-mCD27 mIgG1 NQ, tetra anti-mCD27 mIgG1 NQ). Statistical significance was determined by calculating the AUC and performing one-way ANOVA on AUC values, with significance values indicated in the figure. **D** Tumor area for mice inoculated with B16-OVA cells on day 0 and treated with OVA and the indicated antibodies on day 1. Data points are the mean ± SEM of 12 individual mice (n = 12, isotype mIgG1, anti-mCD27 mIgG1, tetra anti-mCD27 mIgG1) across 2 independent experiments or 6 individual mice (n = 6, anti-mCD27 mIgG1 NQ, tetra anti-mCD27 mIgG1 NQ) from 1 experiment. Statistical significance by one-way ANOVA was performed on AUC, with significance values indicated in the figure. Responses of "NQ" antibody variants were not statistically significant compared to the isotype control. Source data are provided as a Source data file.

microscopy (Supplementary Fig. 6B). We also assessed whether the modified structure of the tetravalent anti-hCD27 antibody affected its thermal stability. Although the melting temperature was slightly reduced compared to the bivalent format (56.7 °C vs. 63.5 °C; Supplementary Table 1), this is still well above physiological temperatures and the standard accelerated storage temperature of 40 °C for biopharmaceuticals, including antibodies[38].

SPR analysis revealed that tetravalent anti-hCD27 mAb demonstrated enhanced avidity for hCD27, as evidenced by a ~2-fold increase in apparent affinity and slower dissociation kinetics compared to bivalent anti-hCD27 (Supplementary Fig. 7A and Supplementary Table 2). Furthermore, the increased avidity of tetravalent anti-hCD27 antibodies was also apparent when binding to hCD27⁺ Jurkat cells was analyzed by flow cytometry, with more tetravalent anti-hCD27 antibody detected at lower concentrations (Supplementary Fig. 7B). The agonistic potential of these anti-hCD27 antibodies was then assessed

using hCD27⁺ Jurkat NF-κB GFP reporter cells. While bivalent anti-hCD27 antibodies (hIgG1, NA, and V11) showed limited activity in the absence of FcγR crosslinking (Fig. 6A, top panels), tetravalent anti-hCD27 antibodies were approximately 1000-fold more potent at inducing NF-κB activation than their bivalent counterparts, even in the absence of FcγRIIB (Fig. 6A, top panels). Although both bivalent and tetravalent antibodies (hIgG1 and V11) showed enhanced activity in the presence of FcγRIIB⁺ CHO cells, tetravalent anti-hCD27 antibodies demonstrated a significant potency advantage, ranging from 40- to 400-fold, compared to their bivalent counterparts, as evidenced by their lower EC50 values (Fig. 6A, lower panels). We then sought to determine if tetravalency and physiological expression of FcγRIIB in peripheral blood mononuclear cells synergized in promoting T cell proliferation. Interestingly, the addition of bivalent and, to a lesser extent, tetravalent anti-hCD27 hIgG1 had a slightly detrimental effect on cell proliferation, likely due to T cell deletion through antibody

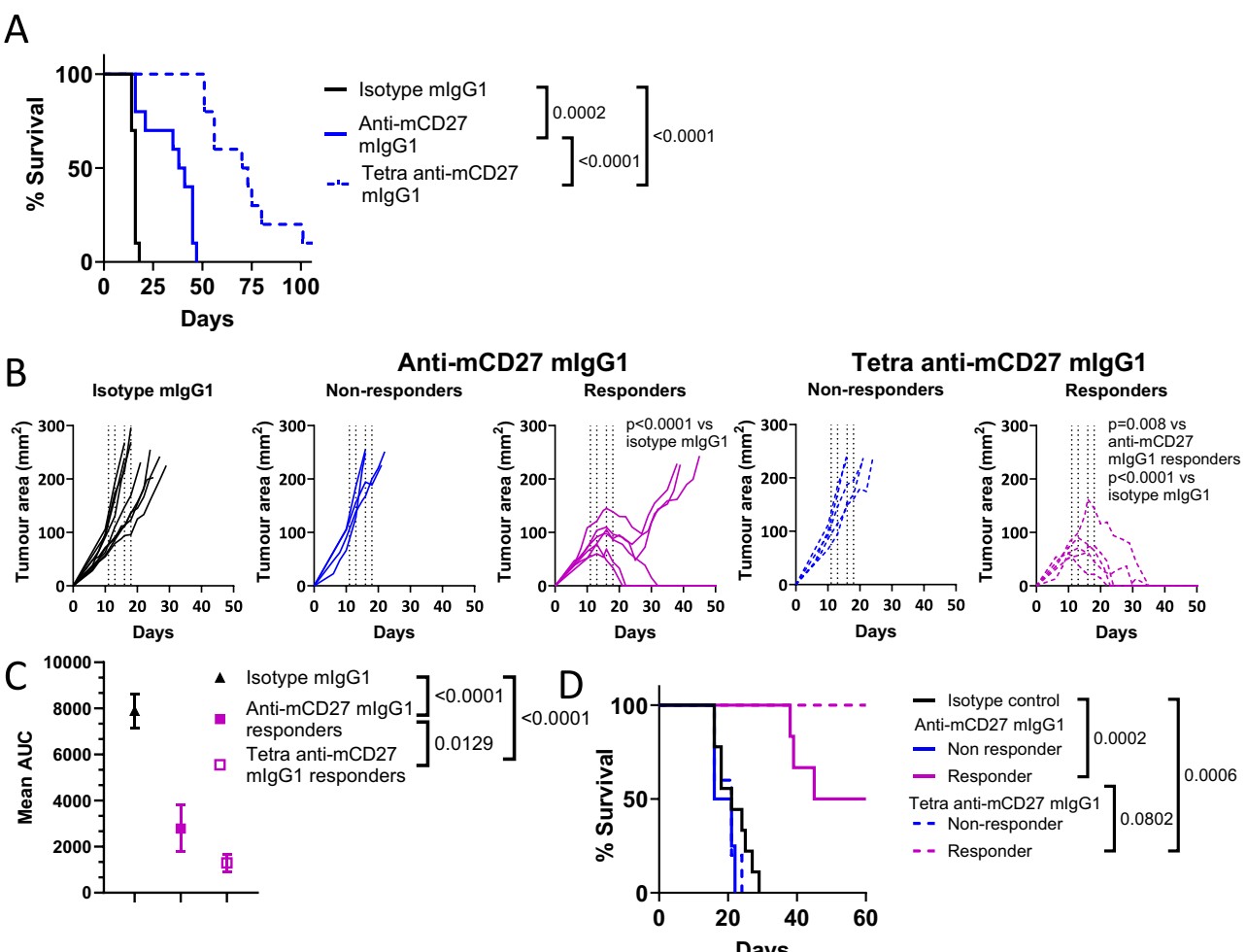

**Fig. 5 | Tetravalent anti-mCD27 generates superior anti-tumor responses compared to its bivalent counterpart. A** Percentage survival to the humane endpoint of Balb/c mice inoculated with BCL$_1$ lymphoma and treated with the indicated antibodies on day 4. Data shown are from 10 mice across 2 independent experiments (n = 10 mice per group). **B** Individual tumor growth curves of Balb/c mice inoculated with CT26 colon carcinoma and treated with the indicated antibodies on days 11, 13, 16, 18. Data shown are from 9 mice (Isotype mIgG1) or 10 mice (anti-mCD27 mIgG1 and tetra anti-mCD27 mIgG1) across 2 independent experiments, with non-responders shown in blue and responders shown in purple.

**C** Summary of tumor growth data from responding and isotype control mice shown in (**B**), calculated as the group mean AUC of Isotype mIgG1 (n = 9), anti-mCD27 mIgG1 (n = 6), or tetra anti-mCD27 mIgG1 (n = 5), with error bars indicating SD. **D** Percentage survival of CT26-bearing mice in (**B**) to the humane endpoint. Statistical significance was determined by Log-rank (Mantel–Cox) test (**A**, **D**), two-way ANOVA (**B**), or one-way ANOVA (**C**), with significance values indicated in the figure. For analysis of statistical significance by ANOVA, Tukey's post-hoc test was used for multiple comparisons. Source data are provided as a Source data file.

engagement of activating FcγRs on blood mononuclear cells (Fig. 6B, left panels). In contrast, both bivalent and tetravalent anti-hCD27 V11 antibodies were able to enhance T cell proliferation, with the tetravalent format demonstrating superior potency, as reflected by lower EC50 values (Fig. 6B, right panels). Moreover, the importance of FcγRIIB engagement was further substantiated by the loss of costimulatory activity of the tetravalent anti-hCD27 NA variant (Fig. 6B, center panels). Thus, the synergistic effect of antibody tetravalency and FcγRIIB engagement is also evident in the human setting.

### Tetravalency enhances hCD27 clustering, while FcγRIIB engagement promotes polarization and reduces internalization

The recently resolved structure of the CD27-CD27 ligand (CD70) complex, which comprises 3 receptor molecules bound to homotrimeric CD70, suggests that CD27 signaling is initiated through receptor clustering[39]. Moreover, ligand-bound CD27 superclusters with a higher receptor density may form through disulfide-bond dimerization of CD27 in the cell membrane[39]. To explore the effects of antibody valency on the distribution of membrane-anchored CD27, we

cultured Jurkat cells expressing hCD27-GFP fusion protein with either bivalent or tetravalent anti-hCD27 hIgG1 V11 and visualized the distribution of receptors by confocal microscopy. Initially, we assessed antibody-induced receptor clustering in the absence of FcγRs and showed that the increased valency afforded by tetravalent anti-hCD27 mAb induced significantly more clusters per cell than bivalent anti-hCD27 at concentrations from 0.1 to 100 nM (Fig. 7A, B). Since the functional activity of tetravalent anti-hCD27 hIgG1 V11 was improved by FcγRIIB binding, we sought to understand how this affects CD27 distribution in the membrane. Therefore, hCD27-GFP$^+$ Jurkat cells were co-cultured with either hFcγRIIB$^+$ CHO cells or control CHO cells together with the tetravalent anti-hCD27 hIgG1 V11 antibody. Whereas CD27 clusters formed in the absence of FcγRIIB were randomly distributed and rapidly internalized within 20–30 min, those formed by tetravalent anti-hCD27 hIgG1 V11 in the presence of hFcγRIIB$^+$ CHO cells were localized to cell-cell contacts (Fig. 7C) and persisted on the cell surface (Fig. 7C, D). Finally, we addressed if hyper-crosslinking of tetravalent anti-hCD27 mAb with a secondary antibody recapitulates the stimulatory effects of FcγRIIB engagement. Using hCD27$^+$ Jurkat

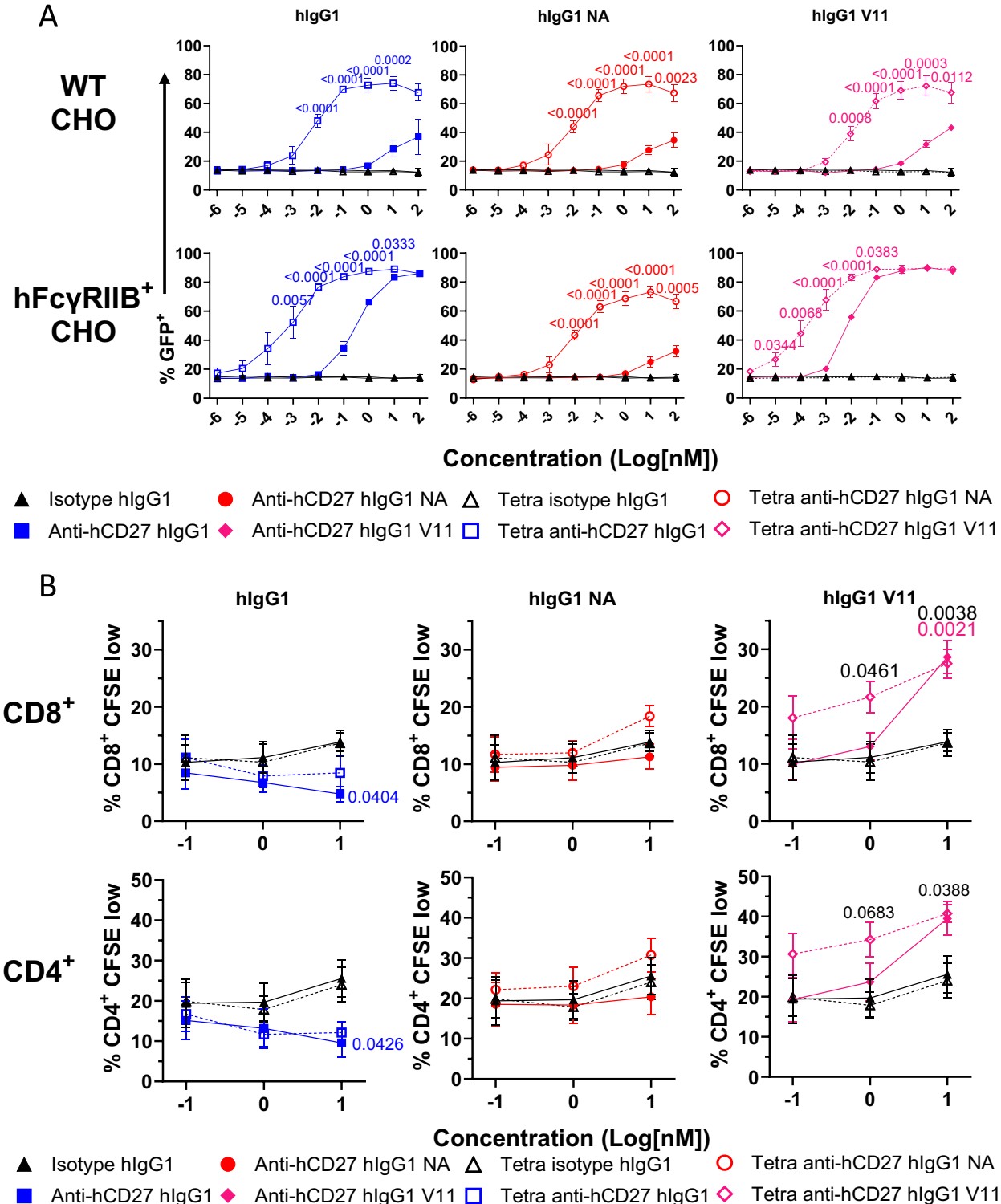

**Fig. 6 | Tetravalency and FcγRIIB engagement synergize for optimal stimulation of human CD27. A** hCD27+ Jurkat NF-κB GFP reporter cells were co-cultured with either WT CHO or hFcγRIIB+ CHO cells in the presence of the indicated antibodies. NF-κB activation was quantitated by GFP expression using flow cytometry. GFP production from Jurkat cells after co-culture with WT CHO cells (above) or hFcγRIIB+ CHO cells (below) and the specified antibodies; hIgG1 (left), hIgG1 NA (center), or hIgG1 V11 (right). Data points are the mean ± SEM (n = 3) from 3 independent experiments. Statistical significance was determined by one-way ANOVA (Tetra anti-hCD27 versus anti-CD27 IgG), with significance values indicated in the figure. **B** Human CFSE-labeled PBMCs were stimulated with sub-optimal anti-CD3

and the indicated antibodies, hIgG1 (left), hIgG1 NA (center), or hIgG1 V11 (right). CD8+ (above) and CD4+ (below) T cell proliferation was assessed by CFSE dilution, with CFSE low cells identified by comparison to cells that were cultured in the absence of any stimulation. Each data point is the mean ± SEM (n = 5) from 5 independent experiments across 5 different donors. Statistical significance between bivalent anti-CD27 and bivalent isotype control (colored numbers) and tetravalent anti-CD27 and tetravalent isotype control (black numbers) was determined by one-way ANOVA, with significance values indicated in the figure. Source data are provided as a Source data file.

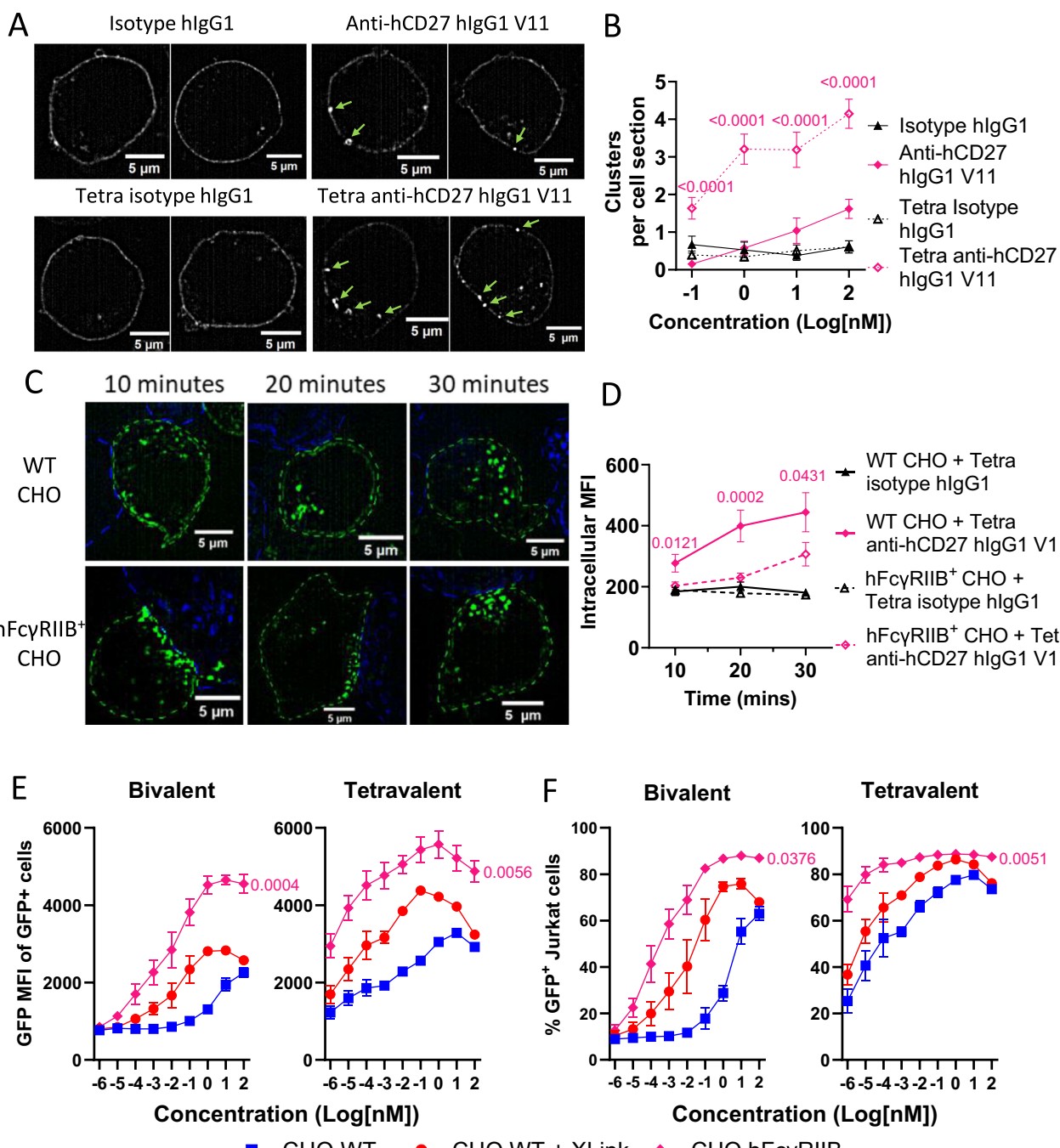

**Fig. 7 | Tetravalent anti-hCD27 hIgG1 V11 induces greater cluster formation, with FcγRIIB engagement polarizing hCD27 clusters and reducing internalization. A, B** Jurkat cells expressing hCD27-GFP⁺ fusion protein were stimulated with the specified antibodies at the indicated concentrations, before PFA fixation and confocal imaging. **A** Representative images at 100 nM (white corresponds to hCD27-GFP fluorescence; clustered areas of the signal are indicated by green arrows) and **B** quantification of the number of clusters per cell section, performed using ImageJ. Data shown are mean ± SEM of CD27 clusters per cell section from 2 independent experiments (n = 54 cells). Statistical significance was determined by one-way ANOVA, with significance values indicated in the figure. **C, D** hCD27-GFP⁺ Jurkat cells were stimulated with 1 nM tetra anti-hCD27 hIgG1 V11 and either WT CHO cells or hFcγRIIB⁺ CHO cells for the specified time points before PFA fixation and confocal imaging using Oxford Nanoimager. **C** Representative images of hCD27-GFP⁺ Jurkat cells with WT CHO (upper panels) or hFcγRIIB⁺ CHO (lower panels) for the specified time points. Dashed blue and green lines (determined by

brightfield and confocal images) indicate the plasma membrane of CHO and Jurkat cells, respectively. **D** Quantification of intracellular GFP MFI of individual cell sections of hCD27-GFP⁺ Jurkat cells after stimulation. Data are the mean ± SEM from 2 independent experiments (n = 24 cells). Statistical significance was determined by one-way ANOVA, with significance values indicated in the figure. **E, F** hCD27⁺ Jurkat NF-κB GFP reporter cells were stimulated with the indicated concentrations of either bivalent anti-hCD27 hIgG1 V11 (left panel) or Tetra anti-hCD27 hIgG1 V11 (right panel) in the presence of WT CHO cells (WT CHO), WT CHO cells and a goat anti-hFc f(ab)₂ (WT CHO + Xlink) or hFcγRIIB⁺ CHO cells (hFcγRIIB⁺ CHO). GFP reporter production was determined by flow cytometry and presented as **E** mean fluorescence intensity (MFI) of GFP⁺ cells and **F** percentage of GFP⁺ cells· Data points are the mean ± SEM (n = 3) from 3 independent experiments. Statistical significance between hFcγRIIB⁺ CHO and WT CHO + Xlink was determined by one-way ANOVA on AUC values, with significance values indicated in the figure. Source data are provided as a Source data file.

NF-κB GFP reporter T cells, we found that hyper-crosslinking increased the stimulatory effects of soluble tetravalent anti-hCD27, as evidenced by the higher GFP signal and increased proportion of GFP⁺ Jurkat cells (Fig. 7E, F). Importantly, however, engagement of FcγRIIB by tetravalent anti-hCD27 provided even greater levels of GFP (Fig. 7E, F), demonstrating that capture of tetravalent anti-hCD27 by FcγRIIB-expressing cells synergizes with antibody tetravalency to maximize CD27 signaling.

## Discussion

The importance of the CD27-CD70 costimulatory pathway in human immunity is underscored by the immune dysregulation observed in individuals with biallelic mutations in the CD27 and CD70 genes that lead to the absence of protein expression[40–43]. These individuals often experience uncontrolled Epstein-Barr virus infection, severe infectious mononucleosis, lymphoproliferation and lymphoma, concomitant with impaired generation of memory B and T cells and defective effector CD8⁺ T cell function. Moreover, studies in mice have shown that CD27 agonism can augment anti-tumor CD8⁺ T cell responses when delivered in the form of monotherapy or in combination with immune checkpoint inhibition[19,20,22]. Despite the promising preclinical data, the clinical response to Varlilumab, a human IgG1 anti-CD27 antibody, has been underwhelming[25,26], prompting further research to elucidate the underlying reasons. The archetypal TNFSF/TNFRSF complex organization, consisting of a trimeric ligand engaging 3 receptors, necessitates appropriate consideration in the design of therapeutic agonists. Although soluble recombinant CD70 fusion proteins that retain activity have been produced, their development as drug modalities is confounded by the need to engineer highly oligomeric proteins, which typically exhibit low production yield and poor pharmacokinetics[33]. The classical bivalent nature of antibodies makes them sub-optimal as TNFRSF agonists, and except for some rare antibodies that exhibit intrinsic agonistic activity, the majority are poor agonists without further engineering[7]. One approach to overcome this limitation is through inhibitory FcγRIIB-mediated antibody crosslinking, which is thought to mimic receptor oligomerization induced by membrane-anchored TNFSF ligands[8,9,13]. Alternatively, antibody engineering strategies, such as modulating affinity[10], promoting Fab-Fab[16] or Fc-Fc[44] interactions as well as restricting Fab conformational flexibility[17,18,45], have been explored to enhance the activity of soluble anti-TNFRSF antibodies. However, the extent to which these approaches can replace FcγRIIB-dependent crosslinking remains unclear. In this study, we sought to address the role of valency and FcγR binding in driving agonism by anti-CD27 antibodies. Consistent with previous reports, we found that the agonistic activity of bivalent anti-mCD27 mIgG1 was dependent on FcγR-mediated crosslinking both in vitro (Fig. 2) and in vivo (Figs. 3 and 4). Although FcγR engagement was still required for agonism by the tetravalent anti-mCD27 antibody, the potency of agonism was significantly improved as determined by measurement of NF-κB activation, primary T cell proliferation, and in vivo expansion of antigen-specific CD8⁺ T cells (Figs. 2–4). Importantly, the enhanced agonistic activity conferred by tetravalency resulted in superior anti-tumor efficacy across three distinct tumor models (Figs. 4 and 5). Notably, this increased therapeutic potency was not associated with observable adverse effects, as indicated by stable body weight measurements in treated mice throughout the study period (Supplementary Fig. 4A), and the absence of elevated liver enzymes (ALT and AST) in the blood (Supplementary Fig. 4B). Furthermore, analysis of tumor-infiltrating lymphocytes in the CT26 model revealed that treatment with tetravalent anti-mCD27 led to an increased frequency of activated CD8⁺ T cells, with no detectable changes in CD4⁺ T cells or regulatory T cells (Supplementary Fig. 5). These findings support the notion that the anti-tumor activity of tetravalent anti-mCD27 involves direct co-stimulation of CD8⁺ T cells. To explore the generalizability of this approach, we produced a tetravalent form of an anti-hCD27 antibody and engineered variants thereof that preferentially bind to different FcγRs. While bivalent anti-hCD27 displayed limited intrinsic agonistic activity in NF-κB reporter assays, the tetravalent format significantly enhanced this activity, regardless of FcγRIIB presence (Fig. 6A). However, in the T cell proliferation assay, the superior potency of tetravalent anti-hCD27 was observed only with the V11 variant, which displays selectively enhanced binding to FcγRIIB[12] (Fig. 6B). In fact, treatment with anti-hCD27 hIgG1, an isotype that exhibits a high activating to inhibitory FcγR binding ratio, resulted in a reduction in the proportion of proliferating T cells, consistent with the strong depletion properties of hIgG1 (Fig. 6B). This finding is reminiscent of the profound decrease in circulating CD4⁺ T cells seen in individuals following administration of anti-hCD27 hIgG1[25,26] and suggest that the unmodified hIgG1 backbone could limit the efficacy of anti-CD27 antibodies. Our combined analysis of antibody binding, imaging, and functional data indicates that the higher avidity and slower dissociation rate of tetravalent anti-CD27 antibodies facilitate efficient receptor clustering and enhanced stability of complexes compared to bivalent antibodies. Subsequent binding of the antibody-receptor complexes to FcγRIIB promotes cluster polarization to the cell-cell interface and reduces CD27 internalization, resulting in an overall increase in the concentration of active receptors at the cell-cell interface. The reduction of CD27 internalization is likely to be significant as CD27 has been shown to undergo clathrin-mediated endocytosis and degradation after binding to its natural ligand, CD70[46]. Furthermore, the inability of secondary antibody crosslinking to fully recapitulate the strong NF-κB activation observed in the presence of FcγRIIB-expressing cells emphasizes the importance of FcγRIIB-mediated CD27 polarization to the cell-cell interface for optimal activation. Although trimerization of some TNFRSF members, such as TNFRI, is thought to be sufficient for transducing downstream signaling, others like CD40, CD30, and CD27 require additional oligomerization[33,47–50]. For example, previous studies have shown that the binding of soluble trimeric CD70 to CD27 drives limited downstream signaling, but this could be enhanced by hexamerization[33]. Interestingly, the activity of a hexameric CD70-Fc dimer of trimer protein was also improved by FcγRIIB binding, suggesting similarities in the mechanisms by which FcγRIIB promotes the activity of CD27 agonists[33]. It remains to be determined whether other receptors can mimic the function of FcγRIIB in enhancing CD27 agonism. Exploring alternative anchoring receptors could provide opportunities to fine-tune CD27 signaling and optimize therapeutic outcomes. For example, targeting receptors that are preferentially expressed in tumor or lymph node microenvironments could lead to activation of different T cell subsets. Given the limited clinical success of current CD27-targeted therapies, our findings highlight the potential of leveraging multivalency and selective FcγR binding to transform anti-CD27 agonists and therefore offer a promising avenue for the development of next-generation anti-CD27 therapies. Moreover, the same approach could be utilized to optimally agonize other members of the TNFRSF, such as OX40, where a lack of activity and disappointing clinical efficacy have been observed[51–55].

Our findings underscore the therapeutic potential of the tetravalent antibody format, which demonstrated robust anti-tumor activity across multiple preclinical tumor models and outperformed its bivalent counterpart. While these results are promising, several considerations remain regarding the translational applicability and developability of higher-valency formats. In our studies, the tetravalent construct showed a modest reduction in plasma half-life and thermal stability relative to the bivalent format, which may warrant further optimization during development. The increased structural complexity may also carry a higher risk of immunogenicity in clinical settings, though this remains to be formally assessed. An important question is whether alternative strategies to promote receptor clustering, such as biparatopic antibodies[56] that engage distinct epitopes

on the same target in trans, could achieve similar functional outcomes without the need for increased valency. These approaches could be evaluated alongside tetravalent formats to optimize the balance between efficacy, pharmacological properties, and manufacturability in future antibody development.

# Methods

## Mice

C57BL/6J (Charles River UK, Strain code: 027), Balb/c (Strain code: 028, Charles River UK), and OT-I transgenic (Tg) mice (Charles River France, strain code: 642) were purchased from Charles River and stock colonies maintained by the University of Southampton Biomedical Research Facility. For C57BL/6 mice, 54 female 8–12-week-old mice were used. For Balb/c mice, 81 female 8–12-week-old mice were used. For OT-I Tg mice, 20 female 8–12-week-old mice were used. All mice were randomly assigned into experimental groups, with experimental and control animals co-housed. Mice were maintained on a 12-h light and dark cycle, an ambient temperature of 20–24 °C, 55% humidity ± 15%, with food and water ad libitum. Mice were kept under specific pathogen-free (SPF) conditions. Daily checks were performed to ensure mice remained healthy, and environmental enrichment was provided. Mice were euthanised by $CO_2$ inhalation or cervical dislocation. All experiments were conducted following University of Southampton ethical approval and in accordance with the Animals (Scientific Procedures) Act 1986 as set out in PPL: P4D9C89EA and PIL: I66C5D543.

## CD45.1 OT-I Tg adoptive transfer

To assess antigen-specific responses in vivo, splenocytes from OT-I Tg CD45.1 mice ($n = 20$) containing $1 \times 10^4$ CD8$^+$ T cells were adoptively transferred into C57BL/6J mice. After 24 h, mice were intravenously challenged with a single dose of 30 nmole OVA$_{257–264}$ peptide and 200 µg of antibody. Antigen-specific T cell expansion was measured by peripheral blood sampling and identifying CD45.1 CD8$^+$ T cells by flow cytometry, at the time points indicated in Fig. 3.

## Tumor models

B16-OVA cells ($2 \times 10^5$) were subcutaneously injected into the flank of C57BL/6J ($n = 48$, 8–12-week-old females) mice on day 0. Twenty-four hours later, mice were intravenously injected with a single dose of 5 mg Ovalbumin (Sigma, A5503) and a molar equivalent quantity (1.33 nmol) of bivalent or tetravalent anti-mCD27 antibody. Antigen-specific T cell expansion was measured by peripheral blood sampling and detection of OVA$_{257–264}$-MHC H2-K$^b$ tetramer$^+$ CD8$^+$ T cells by flow cytometry, at the time points indicated in Fig. 4. Tumor growth was monitored every 2–3 days by calliper, and mice euthanised when cross-sectional area reached 225 mm$^2$. This is below the maximum allowed volume of 4000 mm$^3$, equivalent to 400 mm$^2$. For the BCL$_1$ lymphoma model[19,57], $5 \times 10^6$ tumor cells obtained from in vivo passage were intravenously injected into Balb/c mice ($n = 30$, 8–12-week-old females) on day 0, and a molar equivalent quantity of bivalent or tetravalent anti-mCD27 antibody (0.5 nmol) was injected intravenously on day 4, as a single dose. Tumor growth was monitored every 2–3 days by splenic palpation, with humane endpoint of gross splenomegaly, where the spleen is enlarged and extended, weighing approximately 500–1000 mg. Survival period to the humane endpoint was plotted using the Kaplan–Meier method with analysis for significance by the log-rank (Mantel–Cox) test. CT26 cells ($5 \times 10^5$) were subcutaneously injected into the hind flank of Balb/c mice ($n = 29$, 8–12-week-old females) on day 0, and a molar equivalent of isotype mIgG1, bivalent anti-mCD27 mIgG1, or tetravalent anti-mCD27 mIgG1 (0.66 nmol) was injected on days 10, 13, 16, and 20. Mouse weight was monitored after the first antibody injection. Tumor growth was monitored every 2–3 days by caliper, and mice euthanised when cross-sectional area reached 225 mm$^2$. This is below the maximum allowed volume of 4000 mm$^3$,

equivalent to 400 mm$^2$. Survival period to the humane endpoint was plotted using the Kaplan–Meier method with analysis for significance by the log-rank (Mantel–Cox) test. For all tumor experiments, early termination was carried out if mouse welfare was deemed compromised, in accordance with the guidelines proposed by Foltz and Cullere[58].

## CT26 tumor harvest

CT26 cells ($5 \times 10^5$) were subcutaneously injected into the hind flank of Balb/c mice ($n = 10$, 8–12-week-old females) on day 0, and a molar equivalent of isotype mIgG1 or tetravalent anti-mCD27 mIgG1 (0.66 nmol) was injected on days 10 and 13. Mice were euthanised on day 16, and tumors were excised. Tumors were then diced, incubated with 50 µg/ml DNase I (Roche, 4716728001) and 0.5 WU/ml Liberase DL (Merck, LIBDL-RO) for 30 min at 37 °C. Digestion was inhibited using cRPMI supplemented with 10 mM EDTA (Fisher Scientific, S311-100) and 20 mM HEPES (Gibco, 15630080). Tumor was then passed through a 70 µm cell strainer (Falcon, 352350), and cells were subsequently stained as specified for flow cytometry.

## Antibody bioavailability

200 µg of specified antibody was injected intravenously into C57BL/6J mice ($n = 6$, 8–12-week-old females) on day 0, with blood sampling occurring on days 1, 3, and 7. Serum concentration was determined by ELISA, as previously reported[33]. Briefly, mCD27-ECD-hFc (R&D Systems) was used as a capture reagent, and bound anti-mCD27 antibodies were detected using horseradish peroxidase-conjugated goat anti-mouse IgG.

## Alanine aminotransferase (ALT) and aspartate aminotransferase (AST) detection assay

Balb/c mice ($n = 12$, 8–12-week-old females) were intraperitoneally injected with PBS or 200 µg of bivalent anti-mCD27 mIgG1 or tetravalent anti-mCD27 mIgG1 on day 0. Blood sampling was performed on days 3 and 7, after which samples were allowed to clot overnight at 4 °C. Serum was then assessed for ALT and AST enzymatic activity using an ALT activity assay (Sigma, MAK052) and AST activity assay (Sigma, MAK055), respectively. The activity assays were performed as per the manufacturer's instructions.

## Cell culture

The Jurkat NF-κB reporter T cell line (Jurkat NF-κB GFP, System Biosciences, TR850A-1), CHO-K1 cells (ATCC, CCL-61) and CT26 colon carcinoma (ATCC, CRL-2638) were grown in Roswell Park Memorial Institute (RPMI) medium 1640 (Gibco, 11875093), supplemented with 10% v/v fetal bovine serum (FBS, Sigma, F9665), 2 mM L-Glutamine (Gibco, 25030081), 1 mM Sodium Pyruvate (Gibco, 11360070), 100 U/ml Penicillin (Sigma, P4333-100ML), 100 µg/ml Streptomycin (Sigma, P4333) (cRPMI). Jurkat cells expressing hCD27-GFP and Jurkat NF-κB GFP mCD27 reporter cells were described previously[33,37]. CHO hFcγRIIB (provided by Dr Robert Oldham) and CHO mFcγRII cell lines (provided by Dr Hannah Smith) were cultured in cRPMI containing 1 mg/ml geneticin (Gibco, 10131027). Jurkat NF-κB GFP hCD27 cells[37] were cultured in cRPMI containing 5 µg/ml puromycin (InVivoGen, ant-pr-1). B16-OVA (provided by Dr Caetano Reis e Sousa) were grown in complete Dulbecco's Modified Eagle Medium (DMEM, Gibco), supplemented with 10% v/v FBS, 2 mM L-Glutamine, 1 mM Sodium Pyruvate, 100 U/ml Penicillin, 100 µg/ml Streptomycin.

## Antibodies

The sequences of the anti-mCD27 and anti-hCD27 (hCD27.15) antibodies were derived from the AT124-1 hybridoma[19,33] and US patent 9527916, respectively, and recombinant antibodies were produced as described previously[10,33]. To produce tetravalent antibodies, additional V$_H$-C$_H$1 domains were encoded in the antibody cDNA, linked to the

N-terminus of the heavy chain by a 2(GGGGS) linker. All antibody preparations were endotoxin low (<5 EU endotoxin/mg) as determined using the Endosafe-PTS portable test system (Charles River Laboratories).

## Flow cytometry

Fluorescently labeled antibodies were purchased from eBioscience: allophycocyanin (APC)-labeled or PerCP-Cyanine 5.5-labeled anti-mCD8α (53-6.7), APC-eF780-labeled anti-mCD45.2 (104), R-phycoerythrin (PE)-labeled anti-mFoxP3 (FJK-16s), Fluorescein isothiocyanate (FITC)-labeled anti-mCD4 (GK1.5), eFluor450 (eF450)-labeled anti-mCD45.1 (A20), eF450-labeled anti-hCD3 (UCHT1), eF450-labeled anti-m4-1BB (17B5), eF506-labeled anti-hCD4 (RPA-T4); or Biolegend: PE-Cyanine 7-labeled anti-hCD8α (RPA-T8), Alexa Fluor 647-labeled anti-granzyme B (GB11). OVA$_{257-264}$-specific T cells were identified with PE-labeled H2-K$^b$ OVA$_{257-264}$ tetramer, produced in-house. Fluorescently labeled secondary F(ab)$_2$ antibodies were purchased from Jackson ImmunoResearch: allophycocyanin (APC)-labeled AffiniPure goat anti-mouse Fcγ fragment, APC-labeled AffiniPure goat anti-human Fcγ fragment. If samples contained erythrocytes, these were first lysed using ACK buffer. Viability staining was performed with Fixable Viability Dye eFluor 506 (eBioscience) as per the manufacturer's instructions. FcγR-blocking was then performed with either 10 µg/ml anti-FcγRII/III (2.4G2, produced in-house) or 10% human AB serum (for human cells), at 4 °C for 10 min, prior to a 30 min, 4 °C incubation with surface staining antibodies. If secondary detection with F(ab)$_2$ was used, the cells were washed after surface staining and then incubated with the secondary F(ab)$_2$ for 20 min at 4 °C. For intracellular staining, cells were fixed and permeabilised using the FoxP3/Transcription Factor Staining Kit, as per the manufacturer's instructions (eBioscience, 00-5523-00). All antibodies and dilutions used are listed in Supplementary Tables 3 and 4. Cells were then washed before analysis on a BD FACS Canto II or a BD FACS Calibur using the BD FACSDiva software or BD CellQuest software, respectively.

## Negative stain electron microscopy (EM)

Glow-discharged carbon-coated copper grids (300 mesh, TAAB Laboratories, C267/050) were floated on a droplet of antibody on parafilm, containing 1–5 µg/ml of tetravalent anti-CD27 antibody diluted in H$_2$O, for 1 min. The grids were then briefly floated on a drop of H$_2$O and then a drop of 1% w/v uranyl acetate. Finally, the grids were floated on 1% uranyl acetate (Electron Microscopy Sciences, 22400) for 30 s before blotting on filter paper and air-drying overnight. Images were captured using a transmission electron microscope (Hitachi HT7700, Voltage: 100 kV, magnification: 70,000). Class averages were generated using RELION v3.1.3 (Sjors Scheres, MRC Laboratory of Molecular Biology). In brief, micrographs were uploaded to RELION, which auto-selected particles in a template-free manner using the Laplacian of Gaussian method. Particles then underwent 2D classification into class averages, with averages selected that were well resolved and contained real particles. 2D classification was repeated to refine class averages into recognizable tetravalent antibodies, with representative class averages selected.

## SPR

The apparent affinity of antibodies for their target proteins was analyzed using a Biacore T200 (Biacore). The target protein was captured using an anti-hFc or anti-his antibody that had been amine-coupled to a CM5 chip (Cytiva, 29104988), as per the manufacturer's instructions. Target proteins were mCD27-ECD-hFc (R&D Systems, 574-CD-050) or hCD27 ECD-Fc-his (R&D Systems, 382-CD-100), as indicated in Fig. 1 and Supplementary Fig. 7. The amount captured was optimized for each receptor. Antibodies diluted in HBS-EP+ (Cytiva, BR100669) to the concentrations indicated in Fig. 1 and Supplementary Fig. 7 were

injected into the flow cell at 30 µl/min for 300 s, to allow the antibodies to associate with the target protein. HBS-EP+ buffer was then injected into the flow cells at 30 µl/min to allow the antibody to dissociate, before the chip was regenerated with 3 M MgCl$_2$ at 20 µl/s for 1 min. Kinetics were then assessed in the Biacore Kinetics v3 software.

## SDS-PAGE and SEC

Protein samples (1–2 µg) were incubated with reducing Lammeli buffer containing additional SDS (62.5 mM Tris pH 6.5 (Sigma, 20–160), 4% w/v SDS (VWR chemicals, 44215HN), 10% v/v glycerol (Sigma, G9012), 0.04% v/v bromophenol blue (VWR Chemicals, BDH7392-2) and 62.5 mM DTT (R0861)) for 30 min at ambient temperature. Protein samples were boiled for 5 min and then loaded onto a 10% Bolt Bis-Tris Gel (Invitrogen, NW00100BOX), and separation was performed at 100 V for 15 min and then 140 V for 1 h. The gel was fixed using 10% acetic acid (VWR Chemicals) and 25% isopropanol (Fisher Scientific, 10173240) and then stained with 60 mg/L Brilliant Blue (Sigma, B0770) in 10% acetic acid. Excess Coomassie Blue staining was removed by multiple washes with 10% acetic acid on a rocker for 30 min to 1 h. The gel was imaged using the UVP BioSpectrum AC Imaging System (UVP). Antibodies (20–40 µg/sample) were assessed by analytical SEC using a Zorbax GF-250 column (Agilent) and a 0.2 M phosphate running buffer pH 7.0 containing 1 M dimethyl formamide, at a flow rate of 0.4 ml/min.

## Nano differential scanning fluorimetry

PBS control and antibodies (1 mg/ml in PBS) were loaded into the Prometheus NT.48 (NanoTemper technologies). The temperature was increased from 14 °C to 95 °C at 1 °C per minute, with intrinsic protein fluorescence collected at 330 nm and 350 nm. Unfolding temperature was calculated by PR ThermControl software (NanoTemper technologies) using the first derivative of the 350/330 nm fluorescence.

## Jurkat NF-κB GFP activation

The Jurkat NF-κB GFP reporter cell lines (System Biosciences) expressing either mCD27[33] or hCD27[37] were described previously. To investigate NF-κB activation, cells were incubated with anti-CD27 antibodies for 5 h at 37 °C and either untransfected, mFcγRII-transfected (provided by Dr Hannah Smith), or hFcγRIIB-transfected CHO cells (provided by Dr Robert Oldham). GFP production was detected by flow cytometry.

## Mouse T cell proliferation

Splenocytes $2 \times 10^5$/well from OT-I Tg C57BL/6 mice were stimulated with 1 pM OVA$_{257-264}$ peptide (Peptide Synthetics, 47874) and anti-mCD27 antibodies or an isotype control. Cells were incubated at 37 °C, 5% CO$_2$ for 48 h before 1 µCi of $^3$H-thymidine was added to each well for 16 h. Cells were then lysed, with lysates transferred to filter plates (Opti-plate-96, Perkin Elmer, 101687-172). 40 µl/well Scintillant fluid (Perkin Elmer) was then added before incorporation of $^3$H-thymidine was measured by scintillation counting (TopCount).

## Human T cell proliferation

Human PBMCs were isolated from anonymised leukocyte cones from healthy adult donors through the NHS blood and transplant service. The use of human tissue was approved by the East of Scotland Research Ethics Service, Tayside, UK, and via the Faculty of Medicine Research Ethics Committee under submission 19660. PBMCs were isolated from leukocyte cones by density gradient centrifugation using Lymphoprep (Stemcell, 18060). Isolated PBMCs were labeled with 2 µM CFSE (ThermoFisher Scientific, C34554) for 10 min. CFSE-labeled PBMCs $1 \times 10^5$/well were stimulated with 0.1 ng/ml anti-hCD3 (OKT3) and 0.1–10 nM of anti-hCD27 antibodies or isotype control as indicated in Fig. 6. CFSE dilution in CD4$^+$ and CD8$^+$ T cells was measured after 96 h by flow cytometry. CFSE low cells were identified by comparison

to CFSE-labeled cells that were cultured in the absence of any stimulation.

## Clustering confocal microscopy

Jurkat cells ($1.5 \times 10^5$) expressing hCD27-GFP fusion protein[37] were stimulated with either bivalent or tetravalent anti-hCD27 antibodies at the concentrations indicated in Fig. 7 in a final volume of 200 µl in a 96-well U-bottomed plate at 37 °C for 10 min. The cells were then fixed with 4% PFA for 20 min before imaging on a poly-D-lysine (Gibco, A3890401) coated µ-slide 18-well microscopy chamber (Ibidi, 81811). Cells were imaged in confocal mode (exposure: 3000 ms; line spacing: 2; 100× objective lens) using an ONI Nanoimager (ONI), before analysis with NimOS v1.18 and ImageJ. Clusters were defined as regions of membrane that had a fluorescence intensity above 15,000, which was determined using the threshold setting on ImageJ (ImageJ).

## Cell-cell clustering

Untransfected or hFcγRIIB+ CHO cells ($10^4$) were labeled with Cell-Tracer far red (Invitrogen, C34564), as per the manufacturer's instructions, and the cells were plated onto a poly-D-lysine-coated µ-slide 18-well microscopy chamber and incubated at 37 °C in 5% $CO_2$ overnight. Jurkat hCD27-GFP cells ($3 \times 10^4$) and antibodies (1 nM) were added to the wells for 10–30 min (as indicated in Fig. 7) before fixing with 4% PFA (ThermoFisher Scientific, 28908) for 20 min. The cells were imaged in confocal mode (488 laser: exposure: 3000 ms, line spacing: 2; 647 laser: exposure: 150 ms, line spacing: 2; 100× objective lens) using an ONI Nanoimager, before analysis with NimOS v1.18 and ImageJ. Internalization was measured by selecting the intracellular region of the Jurkat hCD27-GFP cells as a region of interest (ROI) on ImageJ and then calculating the mean fluorescence intensity of the ROI.

## Statistical analysis

GraphPad Prism 9 was used for statistical analysis. Ordinary one-way analysis of variance (ANOVA) with Tukey's post-hoc multiple comparison test or a paired-t test was used as indicated in the figure legends throughout. Tumor area was analyzed by calculating the area under the curve (AUC) before performing a one-way ANOVA. Where error bars are shown, they indicate SEM or SD as detailed in the figure legends. Survival analysis was performed using log-rank (Mantel–Cox) test between indicated groups. $*p < 0.05$, $**p < 0.01$, $***p < 0.001$, $****p < 0.0001$.

## Reporting summary

Further information on research design is available in the Nature Portfolio Reporting Summary linked to this article.

# Data availability

All data are included in the Supplementary Information or available from the authors, as are unique reagents used in this article. The raw numbers for charts and graphs are available in the Source data file whenever possible. Source data are provided with this paper.

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

## Acknowledgements

The authors thank Dr Hannah Smith and Dr Robert Oldham for the kind gift of CHO cells expressing FcγRs. The authors would like to acknowledge that the purchase of the Nanoimager used in the microscopy studies was made possible by a kind gift from the Mark Benevolent Fund. The authors are grateful to members of the Biomedical Research Facility, University of Southampton, for their help with the murine in vivo studies. The study was funded by Cancer Research UK Award number DRCDDRPGMApr2020\100005 (A.Al-S., S.A.B., M.S.C.).

## Author contributions

M.A.W. performed and analyzed the experiments with the help of A.P., H.J.M., S.G.B., H.T.C.C., T.I., C.A.P., C.I.M., and S.J. A.Al-S. conceived the project. A.Al-S., S.A.B., and M.S.C. supervised the project. M.A.W. and A.Al-S. wrote the manuscript with feedback from M.S.C., S.A.B., and S.H.L.

## Competing interests

A.Al-S. and S.H.L. are inventors on patents pertaining to the generation and therapeutic use of agonist anti-CD27 antibodies and have received research funding from Celldex Therapeutics in relation to anti-CD27 antibodies. The remaining authors declare no competing interests.
