## [Transparent Peer Review file · Nature Communications]

Harnessing Multivalency and FcγRIIB Engagement to Augment Anti-CD27 Immunotherapy

Corresponding Author: Professor Aymen Al-Shamkhani

Version 0:

Reviewer comments:

Reviewer #1

(Remarks to the Author)

The authors convincingly demonstrate the importance of FcGR2B engagement for optimal CD27 agonism in both mouse and human systems. They demonstrate that increasing the number of CD27 binding sites per antibody from two (bivalent) to four (tetravalent) allows for equivalent agonism to occur at significantly lower antibody concentrations and that this is the result of an increased avidity effect driven by a lower antibody off-rate.

In vivo, tetravalent anti-CD27 antibodies outperformed bivalent ones in promoting antigen-stimulated proliferation of adoptively transferred congenic OT-1 cells and in reducing the growth rate of OVA-expressing B16 melanoma cells in ovalbumin-vaccinated mice. Fc domain silencing abrogated the effect, highlighting the importance of FcGR engagement. Additionally, treatment with tetravalent anti-CD27 antibodies also resulted in a longer survival time in mice challenged with the BCL1 lymphoma cell line compared with bivalent anti-CD27 antibodies.

Finally, the authors find that modifying human IgG1 Fc domains to increase FcGR2B and decrease FcGR1 binding strongly enhances the capacity of anti-CD27 antibodies to stimulate proliferation of T-cells in human PBMCs cultured ex vivo. These data suggest that CD27 antibodies with wildtype IgG1 Fc domains induce some level of depletion of T-cells, presumably via phagocytosis or ADCC. This finding provides guidance for the design of new clinical TNFR agonist antibodies.

The results represent an incremental advance in our understanding of optimal agonists and are not highly novel or unexpected. The manuscript is more appropriate for a journal such as *Journal of Immunology* or *mAbs*.

1. The mechanism of the anti-lymphoma effect (BCL1 lymphoma model) should be further explored to understand whether or not this approach is applicable only to lymphoma. Many lymphomas themselves express FcGR2B on their surface, whereas the vast majority of other tumor types do not. Considering the apparent importance of FcGR2B for the activity of anti-CD27 antibodies, an alternate explanation for the anti-lymphoma activity observed by the authors is that the anti-CD27 antibodies bound FcGR2B on the surface of the BCL1 lymphoma cells and directly provided co-stimulation to T-cells interacting with these lymphoma cells. If this is the mechanism, it would be relevant only for FcGR-expressing tumors, which are primarily lymphomas, and would not be applicable to most other malignancies. Treatment of other tumor types would require that CD27 agonism delivered to T-cells interacting with professional APCs (which do express FcGR) is sufficient to drive an anti-tumor response, since the tumor cells themselves would not express FcGR. If BCL1 lymphoma cells do express FcGR2B, then the authors' experiment does not test this. Determine whether the BCL1 lymphoma expresses surface FcGR2B by flow cytometry. This line of reasoning should be discussed in the text so that the reader can draw appropriate conclusions regarding the breadth of potential translatability.

2. Figure 2A shows that Fc receptor expression on the CHO cells is required to induce NFκB agonism in co-cultured anti-murine CD27 mab and murine CD27+ Jurkat cells. However, Figure 5A shows that Fc receptor expression on the CHO cells is not required for agonism of anti-human CD27 mab and human CD27+ Jurkat cells. The reason for this discrepancy is not clear and is not explored. Is it an antibody-specific strength of agonist activity? Discuss.

Minor.

3. The results presented in Figures 6A & 6B may perplex some readers. These results indicate that the anti-CD27 antibody with a modified Fc domain that preferentially binds FcGR2B induced more CD27 clustering on Jurkat cells than the anti-CD27 antibody with a modified IgG1 Fc domain. Add an explanation that no FcR is present in the system and so only the valency is being compared.

4. In Fig1A, highlight the CDR regions.

5. In Fig 2C, put in legend how many days cells were cultured before analysis.
6. In Fig 2C, label x-axis as antibody or peptide concentration to avoid confusion.
7. The NQ mutants have a greater level of activity than control. Saying they are inactive seems an overstatement. Indicate on graphs as NS (not significant) if that is the case.
8. Starting B16-OVA antibody therapy on day 1 after implantation is very early. One is not sure if the tumor has been established and exhaustion has not been established. Note in text these limitations. Have the authors tested at later times?
9. If antibody treatments are only on day one, say a single dose, and if repeated, say so.
10. "lacking a known tumor-specific antigen" change to an "identified tumor-specific antigen"
11. "hCD27.15 [34] in a standard bivalent IgG1 format" human or mouse?
12. If the hCD27.15 mab is the parent of a mab in clinical trials, say the name of the drug.

Reviewer #2

(Remarks to the Author)

Al-Shamkhani and colleagues generated tetravalent versions (Fab on Fab) of anti-CD27 antibodies and studied their potency and agonist activity in vitro and in vivo. CD27 is a costimulatory receptor on T cells and agonistic targeting CD27 is a validated clinical concept. However, the anti-CD27 agonist antibody Varlilumab (human IgG1) displayed limited success. In this paper the authors explored how increase of valency from bivalent to tetravalent can improve potency in conjunction with modulated FcγR binding. The authors confirmed previous work showing that agonistic activity of anti-CD27 antibodies depends on FcγR-mediated crosslinking. Importantly, the authors impressively showed by both in vitro and in vivo studies that the potency of agonism was significantly improved for tetravalent mouse and human derived antibodies. A variety of readouts was performed to corroborate improved potency, which could be related to improved clustering and reduced receptor internalization eventually also resulting in improved tumor responses in two different in vivo tumor models. The paper is well written, the data are clearly presented and sound and statistical significance analysis was performed for each experiment. I recommend publication.

A minor point that might be considered: It is well known that biparatopic antibodies can have beneficial properties regarding receptor clustering (see e.g. <https://www.tandfonline.com/doi/full/10.1080/19420862.2024.2310890>). It might be worth briefly discussing this option in the discussion section (do Varlilumab and hCD27.15 bind distinct epitopes? Ref. 34 is not very informative).

Reviewer #3

(Remarks to the Author)

Marcus A. Widdess and colleagues developed a strategy to improve the activity of CD27 antibody by increasing both antibody valency and FcγRIIB engagement, presenting a promising avenue for the development of next-generation anti-CD27 therapies. Notably, both multivalency and Fc engineering strategies have been explored for OX40 antibodies and agonist antibodies targeting other TNFRSF receptors. To provide more novel and valuable insights in this field, further mechanistic studies are essential. While the authors focused on the CD27 antibody-mediated costimulation effect, the impact of tetravalent CD27 antibodies on Treg cells should also be investigated, given that the depletion of Treg cells by CD27 antibodies is equally crucial to the costimulation effect in clinical trials. Additionally, evaluating the efficacy of tetravalent CD27 antibodies in CD27 humanized mouse models is warranted.

Major points:

- 1) Figure 2. There are different types of FcγRII receptors, such as FcγRIIA, FcγRIIB, and FcγRIIC, which type of receptor is involved in this study?
- 2) CD27 expression on Tregs plays a critical role in limiting immune responses against tumors. While the authors primarily focus on the agonistic properties of the CD27 antibody, it is important to note that CD27 antibodies used in clinical trials are typically of the human IgG1 subclass, which facilitate CD27 crosslinking and mediate Treg depletion via antibody-dependent cellular cytotoxicity. Therefore, to ensure the clinical relevance and translational value of this work, it is essential to investigate both the agonistic effects and the Treg depletion capabilities of the CD27 antibody. Given their enhanced valency, tetravalent CD27 antibodies may exhibit more potent activity in depleting Tregs, which could significantly impact their therapeutic efficacy in cancer immunotherapy.
- 3) Figure 4. To fully evaluate the anti-tumor efficacy of these antibodies, both tumor growth curves and survival curves of the

treated mice should be presented. Additionally, a more detailed analysis of the tumor immune microenvironment is warranted. This should include quantifying different T cell populations (at least conventional T cells [Tcon], regulatory T cells [Treg], and cytotoxic T lymphocytes [CTLs]) and assessing the expression of activation and exhaustion markers on these cells. The current data are presented at a single time point, which provides a snapshot of T cell activation. However, a longitudinal analysis over multiple time points could provide more comprehensive understanding of the dynamics of T cell activation and the sustained effects of the treatment.

4) The toxicity of the tetravalent anti-CD27 antibody has not been assessed. Will the tetravalent antibody have a reduced half-life? The safety and PK data will be critical to the translation of this tetravalent agonist antibody into clinical drug development.

5) There are several clinical candidates, such as varilumab, all of which have shown limited efficacy in both preclinical models and patients. To ensure that this study can potentially be translated into clinical drug development, the author should evaluate the efficacy and toxicity of the tetravalent version of varilumab in a humanised CD27 mouse model. Several companies offer humanised CD27 mice.

Minor points:

1) Table 1, which model is used to fit the SPR data, 1:2 or 1:1 kinetic model? Is the binding affinity apparent K_d ?

2) The authors should use a consistent name throughout the manuscript, for example, CD8+ T cells and CD8 T cells.

3) Figure 4C, the tumour volume at the last time point was less than 200mm³. The tumour growth curve should be plotted until some tumours reach 1500 or 2000mm³.

4) In lines 271-275 of the Methods section, the author should clarify how many doses of antibody were administered. It would be better to inject OVA with an adjuvant to better induce T cell immunity.

6) In the Figure 5 section, the author should discuss that some human CD27 antibodies are less dependent on FcγRIIb cross-linking, but that FcγRIIb cross-linking can further enhance potency, suggesting that the agonistic activity of CD27 antibodies may be influenced by their epitopes. The bivalent antibody and the tetravalent antibody with the Fc V11 variant did not show a significant difference in Figure 5A. What is the reason?

7) In Figure 6C, the video should be provided in supplemental information.

9) For Figure 6E, it would be informative to analyze whether the percentage of GFP+ cells shows significant changes after treatment with either bivalent or tetravalent antibodies. Additionally, to enhance data interpretation, both MFI and the percentage of positive cells could be calculated and presented on the y-axis.

10) The Discussion section would benefit from a comprehensive analysis of the potential limitations associated with the tetravalent antibody approach, including but not limited to its structural stability, pharmacokinetic properties, and potential immunogenicity.

11) The term "Finaly" in line 175 requires correction to "Finally." Additionally, we suggest performing a comprehensive spell-check of the entire manuscript to identify and rectify any similar typo.

12) Although Figure S2 shows no aggregation of the tetravalent antibody, this does not mean that it is stable. The thermostability of the tetravalent antibody should be determined.

Reviewer #4

(Remarks to the Author)

Version 1:

Reviewer comments:

Reviewer #1

(Remarks to the Author)

The authors have done additional experimental work and addressed my concerns.

Reviewer #2

(Remarks to the Author)

In their revision, the authors addressed my critical point adequately. I recommend publication.

Reviewer #3

(Remarks to the Author)

The authors have addressed most of the comments raised during revision. However, several minor points still require clarification before acceptance to Nature Communications.

1. The layout of the main figures requires overall optimization to improve clarity and visual appeal.

2. Although treatment of CT26-bearing mice with the tetravalent antibody did not result in a reduction in body weight, additional safety parameters, such as hepatotoxicity (ALT, AST et al.), should also be assessed to further substantiate the safety profile of the tetravalent antibody.

3. Regarding the response to minor point 9), the figure referred to by the authors as Figure 6F does not appear to

correspond; it may in fact be Figure 7F?

Reviewer #4

(Remarks to the Author)

Point-by-point response

Reviewer #1 (Remarks to the Author):

The authors convincingly demonstrate the importance of FcGR2B engagement for optimal CD27 agonism in both mouse and human systems. They demonstrate that increasing the number of CD27 binding sites per antibody from two (bivalent) to four (tetravalent) allows for equivalent agonism to occur at significantly lower antibody concentrations and that this is the result of an increased avidity effect driven by a lower antibody off-rate.

In vivo, tetravalent anti-CD27 antibodies outperformed bivalent ones in promoting antigen-stimulated proliferation of adoptively transferred congenic OT-1 cells and in reducing the growth rate of OVA-expressing B16 melanoma cells in ovalbumin-vaccinated mice. Fc domain silencing abrogated the effect, highlighting the importance of FcGR engagement. Additionally, treatment with tetravalent anti-CD27 antibodies also resulted in a longer survival time in mice challenged with the BCL1 lymphoma cell line compared with bivalent anti-CD27 antibodies.

Finally, the authors find that modifying human IgG1 Fc domains to increase FcGR2B and decrease FcGR1 binding strongly enhances the capacity of anti-CD27 antibodies to stimulate proliferation of T-cells in human PBMCs cultured ex vivo. These data suggest that CD27 antibodies with wildtype IgG1 Fc domains induce some level of depletion of T-cells, presumably via phagocytosis or ADCC. This finding provides guidance for the design of new clinical TNFR agonist antibodies.

The results represent an incremental advance in our understanding of optimal agonists and are not highly novel or unexpected. The manuscript is more appropriate for a journal such as Journal of Immunology or mAbs.

We thank the reviewer for their comments. Despite the success of checkpoint inhibitors such as anti-PD-1, anti-CTLA-4, and more recently anti-LAG3 (in combination with anti-PD-1), most patients remain refractory to treatment. Therefore, it is critical to explore alternative approaches that could broaden the clinical benefit of immunotherapy.

CD27 is a key driver of CD8⁺ T cell responses, and its non-redundant role in human immunity is underscored by genetic deficiencies that lead to susceptibility to EBV-associated malignancies and other lymphoproliferative disorders, alongside defects in T cell activation, cytotoxicity, and memory formation. However, translating the potent biological effects of CD27 co-stimulation into therapeutics has been challenging. The currently explored anti-CD27 IgG molecules have shown limited efficacy as agonists to date, failing to replicate the effects of the natural, trimeric, membrane-bound CD70 ligand, demonstrating a gap in knowledge as to how to optimally exploit this costimulatory target.

In this manuscript, we present new insight into this issue and an approach to overcome these limitations. Unexpectedly, we found that optimal CD27 agonism requires **both** increased valency, specifically tetra-avalency compared to bivalency, **and** stabilisation of the receptor complex at the plasma membrane via co-engagement with FcγRIIB. We demonstrate, using three distinct in vivo preclinical models (including one added in the revised manuscript), that this strategy represents a significant advancement in the development of effective CD27 agonists.

1. The mechanism of the anti-lymphoma effect (BCL1 lymphoma model) should be further explored to understand whether or not this approach is applicable only to lymphoma. Many lymphomas themselves express FcGR2B on their surface, whereas the vast majority of other tumor types do not. Considering the apparent importance of FcGR2B for the activity of anti-CD27 antibodies, an

alternate explanation for the anti-lymphoma activity observed by the authors is that the anti-CD27 antibodies bound FcGR2B on the surface of the BCL1 lymphoma cells and directly provided co-stimulation to T-cells interacting with these lymphoma cells. If this is the mechanism, it would be relevant only for FcGR-expressing tumors, which are primarily lymphomas, and would not be applicable to most other malignancies. Treatment of other tumor types would require that CD27 agonism delivered to T-cells interacting with professional APCs (which do express FcGR) is sufficient to drive an anti-tumor response, since the tumor cells themselves would not express FcGR. If BCL1 lymphoma cells do express FcGR2B, then the authors' experiment does not test this. Determine whether the BCL1 lymphoma expresses surface FcGR2B by flow cytometry. This line of reasoning should be discussed in the text so that the reader can draw appropriate conclusions regarding the breadth of potential translatability.

We would like to thank the reviewer for their insightful suggestion. FcγR2B expression on the BCL₁ lymphoma is well documented and we have referenced this in the revised manuscript. To evaluate whether treatment with anti-CD27 agonists is effective beyond FcγRIIB-expressing B cell lymphomas or the protein vaccination (OVA)/B16-OVA challenge models, we employed the CT26 colon carcinoma model. These data are presented in new Figure 5B–D. We demonstrate that treatment of CT26-bearing mice with the tetravalent anti-mCD27 mIgG1 led to complete tumour regression in all responding mice. In contrast, only half of the responding mice treated with the bivalent anti-hCD27 mIgG1 survived the tumour challenge. These findings demonstrate that the anti-tumour efficacy of an optimised anti-CD27 agonist is not limited to FcγRIIB-expressing B cell lymphomas but also extends to tumours lacking FcγR expression. The new findings are fully discussed in the Results section of the revised manuscript

2. Figure 2A shows that Fc receptor expression on the CHO cells is required to induce NFκB agonism in co-cultured anti-murine CD27 mAb and murine CD27+ Jurkat cells. However, Figure 5A shows that Fc receptor expression on the CHO cells is not required for agonism of anti-human CD27 mAb and human CD27+ Jurkat cells. The reason for this discrepancy is not clear and is not explored. Is it an antibody-specific strength of agonist activity? Discuss.

We agree with the reviewer's comment that in the human NF-κB reporter assay the requirement for FcγRIIB is not absolute, unlike that observed in the reporter assay of murine CD27. However, even in the human assay we observed a substantial enhancement of NFκB activity in the presence of CHO cells expressing FcγRIIB for both the tetravalent hIgG1 and tetravalent hIgG1 V11 formats but not the Fc 'silent' variants – compare new Fig. 6A lower 'hFcγRIIB⁺ CHO' with upper 'WT CHO' panels. Importantly, we demonstrated that effective co-stimulation of both human and murine primary T cells by the anti-CD27 agonists required FcγR engagement (Fig. 2C for murine anti-CD27 mAb and new Fig. 6B - compare 'Fc silent' hIgG1 NA with hIgG1 V11). Therefore, the overall conclusions that tetravalency and FcγRIIB binding combine for optimal agonism is supported by our data. There are several potential reasons for the variability in dependence on FcγRIIB in the NFκB reporter assay. First, there is variability in the baseline levels of NFκB activity. Second, we cannot exclude differences in antibody epitope or downstream cell signalling in the Jurkat reporter cell lines that could affect the sensitivity and therefore the threshold of NFκB activation by mouse and human CD27 crosslinking. For these reasons we employed additional assays where possible (in vitro primary T cells and in vivo priming) to fully address the dependency on FcγRIIB.

Minor.

3. The results presented in Figures 6A & 6B may perplex some readers. These results indicate that the anti-CD27 antibody with a modified Fc domain that preferentially binds FcGR2B induced more CD27 clustering on Jurkat cells than the anti-CD27 antibody with a modified IgG1 Fc domain. Add an explanation that no FcR is present in the system and so only the valency is being compared.

We thank the reviewer for pointing out this oversight. In these experiments we are only comparing the effect of valency. We have added a sentence to clarify that no FcγR is present in these experiments.

4. In Fig1A, highlight the CDR regions.

We have re-drawn the Figure as suggested. Based on known structures of Fab fragments we have adjusted the position of the linkers and highlighted the relative location of the CDRs.

5. In Fig 2C, put in legend how many days cells were cultured before analysis.

The legend already states that the cells were cultured for 72 hours, with the final 16 hours performed in the presence of tritiated thymidine.

6. In Fig 2C, label x-axis as antibody or peptide concentration to avoid confusion.

We apologise for this omission. The x-axis label has been amended to read 'antibody concentration (Log[nM])'.

7. The NQ mutants have a greater level of activity than control. Saying they are inactive seems an overstatement. Indicate on graphs as NS (not significant) if that is the case.

We would like to thank the reviewer for raising this point. The text has been modified to clarify that there was not a statistically significant increase in activity with the NQ variants. This is now also stated in the Figure legends where applicable.

8. Starting B16-OVA antibody therapy on day 1 after implantation is very early. One is not sure if the tumor has been established and exhaustion has not been established. Note in text these limitations. Have the authors tested at later times?

B16-OVA tumours are highly aggressive and resistant to long-term T cell immunotherapy due to the emergence of OVA loss variants (Kazula et al., Int J Cancer 2011). Therefore, we initiated treatment and vaccination early. We have revised the text to acknowledge the limitations of this model and have added the Kazula et al reference. While we have not tested intervention at later time points in this model, we now include new data (Figure 5B–D) demonstrating the activity of the tetravalent anti-CD27 antibody in an established CT26 colon carcinoma model. In these new experiments CT26 tumours were allowed to establish for 10 days to a size of ~100 mm² prior to treatment. In this setting the tetravalent anti-CD27 produced a cure rate of 100% in responders. We also demonstrated superior activity of the tetravalent antibody in the established BCL1 lymphoma model (Figure 5A). Overall, the anti-tumour effects of the tetravalent anti-mCD27 surpassed that of the bivalent counterpart in 3 distinct tumour models.

9. If antibody treatments are only on day one, say a single dose, and if repeated, say so.

We have added a 'a single dose of' where applicable.

10. "lacking a known tumor-specific antigen" change to an "identified tumor-specific antigen"

We have changed this in the text.

11. "hCD27.15 [34] in a standard bivalent IgG1 format" human or mouse?

This refers to human IgG1 and is now indicated in the text.

12. If the hCD27.15 mab is the parent of an mab in clinical trials, say the name of the drug.

hCD27.15 is one of several anti-human CD27 monoclonal antibodies developed by Aduro Biotech and subsequently licensed to Merck. Ultimately, Merck selected hCD27.131A for clinical development as Boserolimab (MK-5980). We selected hCD27.15 for our studies based on its well-characterized properties. In prior work, we evaluated the binding and functional activity of hCD27.15 in comparison with other anti-CD27 antibodies, including the clinically relevant Varlilumab. Among these, hCD27.15 exhibited the strongest activity (Heckel et al., *Communications Biology*, 2022).

Reviewer #2 (Remarks to the Author):

Al-Shamkhani and colleagues generated tetravalent versions (Fab on Fab) of anti-CD27 antibodies and studied their potency and agonist activity in vitro and in vivo. CD27 is a costimulatory receptor on T cells and agonistic targeting CD27 is a validated clinical concept. However, the anti-CD27 agonist antibody Varlilumab (human IgG1) displayed limited success. In this paper the authors explored how increase of valency from bivalent to tetravalent can improve potency in conjunction with modulated FcγR binding. The authors confirmed previous work showing that agonistic activity of anti-CD27 antibodies depends on FcγR-mediated crosslinking. Importantly, the authors impressively showed by both in vitro and in vivo studies that the potency of agonism was significantly improved for tetravalent mouse and human derived antibodies. A variety of readouts was performed to corroborate improved potency, which could be related to improved clustering and reduced receptor internalization eventually also resulting in improved tumor responses in two different in vivo tumor models. The paper is well written, the data are clearly presented and sound and statistical significance analysis was performed for each experiment. I recommend publication. A minor point that might be considered: It is well known that biparatopic antibodies can have beneficial properties regarding receptor clustering (see e.g. <https://www.tandfonline.com/doi/full/10.1080/19420862.2024.2310890>). It might be worth briefly discussing this option in the discussion section (do Varlilumab and hCD27.15 bind distinct epitopes? Ref. 34 is not very informative).

We thank the reviewer for their positive comments and valuable suggestions. We agree that biparatopic antibodies represent a promising class of molecules for promoting agonism. In response, we have added a final paragraph to the Discussion to address this comment and have included the review article recommended by the reviewer. The study by Heckel et al. (*Communications Biology*, 2022) has now been cited in the Results section to highlight the distinct epitopes recognised by Varlilumab and hCD27.15.

Reviewer #3 (Remarks to the Author):

Marcus A. Widdess and colleagues developed a strategy to improve the activity of CD27 antibody by increasing both antibody valency and FcγRIIB engagement, presenting a promising avenue for the development of next-generation anti-CD27 therapies. Notably, both multivalency and Fc engineering strategies have been explored for OX40 antibodies and agonist antibodies targeting other TNFRSF receptors. To provide more novel and valuable insights in this field, further mechanistic studies are essential. While the authors focused on the CD27 antibody-mediated costimulation effect, the impact of tetravalent CD27 antibodies on Treg cells should also be investigated, given that the depletion of Treg cells by CD27 antibodies is equally crucial to the costimulation effect in clinical trials. Additionally, evaluating the efficacy of tetravalent CD27 antibodies in CD27 humanized mouse models is warranted.

We would like to thank the reviewer for their constructive comments, and we are grateful for their insightful suggestions to improve the manuscript. As requested, we have carried out several additional experiments as detailed in the revised manuscript and the point-by-point response below.

Major points:

1) Figure 2. There are different types of FcγRII receptors, such as FcγRIIA, FcγRIIB, and FcγRIIC, which type of receptor is involved in this study?

To clarify, Figure 2 refers to anti-mouse CD27 mIgG1 antibodies. Since mice express only a single isoform of FcγRII, it is typically referred to as FcγRII rather than FcγRIIB, as is the case in the human system where multiple FcγRII isoforms exist. Mouse FcγRII is the sole inhibitory FcγR, and the homolog of human FcγRIIB. For these reasons, we believe the use of the nomenclature "mouse FcγRII" (mFcγRII) in the manuscript is appropriate.

2) CD27 expression on Tregs plays a critical role in limiting immune responses against tumors. While the authors primarily focus on the agonistic properties of the CD27 antibody, it is important to note that CD27 antibodies used in clinical trials are typically of the human IgG1 subclass, which facilitate CD27 crosslinking and mediate Treg depletion via antibody-dependent cellular cytotoxicity. Therefore, to ensure the clinical relevance and translational value of this work, it is essential to investigate both the agonistic effects and the Treg depletion capabilities of the CD27 antibody. Given their enhanced valency, tetravalent CD27 antibodies may exhibit more potent activity in depleting Tregs, which could significantly impact their therapeutic efficacy in cancer immunotherapy.

The reviewer raises an interesting point, which is addressed in our original manuscript. As shown in Figure 6B (originally Figure 5B) there is a dose dependent reduction in activated CD4 and CD8 T cells with anti-human CD27 hIgG1, but this reduction was not potentiated by the tetravalent format of the antibody. Thus, we can conclude that the tetravalent hIgG1 mAb is not more potent at inducing T cell deletion than its bivalent counterpart. Importantly, this data also highlights a caveat in clinical studies that used hIgG1, namely that anti-CD27 hIgG1 deletes both effector T cells as well as Tregs. However, when we converted the antibody to a V11 version, a format with a modified Fc that reduces affinity for activating FcγRs and increases affinity for hFcγRIIB, T cell deletion was abolished and replaced with enhanced proliferation, consistent with enhanced agonism and CD27 signalling (Figure 6B). This highlights the importance of selecting a non-depleting format in order to achieve effective agonism.

In our revised manuscript, we also further addressed this question in the CT26 mouse model using the tetravalent anti-mCD27 antibody with a mouse IgG1 Fc domain. This isotype preferentially engages the inhibitory mFcγRII while showing minimal-to-no binding to activating mFcγRs, mirroring the FcγR binding profile of the human IgG1 V11 variant. In this setting, we observed a marked increase in tumour-infiltrating CD8⁺ T cells, with no corresponding change in Tregs (new Supplemental Figure 5). These findings suggest that the mechanism of action of the tetravalent anti-mCD27 antibody is not dependent on Treg depletion but rather involves selective enhancement of CD8⁺ T cell activation and responses (new Supplemental Figure 5).

3) Figure 4. To fully evaluate the anti-tumor efficacy of these antibodies, both tumor growth curves and survival curves of the treated mice should be presented. Additionally, a more detailed analysis of the tumor immune microenvironment is warranted. This should include quantifying different T cell populations (at least conventional T cells [Tcon], regulatory T cells [Treg], and cytotoxic T lymphocytes [CTLs]) and assessing the expression of activation and exhaustion markers on these cells. The current data are presented at a single time point, which provides a snapshot of T cell activation. However, a longitudinal analysis over multiple time points could provide more

comprehensive understanding of the dynamics of T cell activation and the sustained effects of the treatment.

We thank the reviewer for this suggestion. B16-OVA is known to be resistant to long-term T cell immunotherapy due to the emergence of OVA loss variants (Kaluza et al Int J Cancer 2011). Accordingly, in the B16-OVA model shown in Figure 4, the OVA-specific CD8⁺ T cell response induced by OVA vaccination and anti-CD27 agonist treatment leads only to a transient delay in tumour growth, without a significant improvement in long-term survival. We have amended the manuscript text to acknowledge this limitation and added survival data from a B16-OVA experiment in new Supplemental Figure 3. To complement this data and to address the reviewer's point, we have added a longitudinal analysis of antigen specific T cells (Figure 4C). This new data highlights the longevity of the antigen-specific T cell response elicited by the tetravalent anti-mCD27 antibody. The new Figure 4C complements the original graph which showed the antigen-specific response only at the peak of the response.

The second well-established tumour model we present, the BCL₁ lymphoma (references 19, 35 and 58), primarily grows in the spleen, which limits the ability to directly measure tumour burden. Unfortunately, it is not possible to use fluorescent or luminescent proteins to monitor tumour load because the BCL₁ lymphoma does not grow in vitro and is propagated by in vivo passage and therefore manipulation in vitro eg. transfection to express fluorescent or luminescent proteins cannot be performed. The BCL₁ is a well-established model and the methods to measure survival of BCL₁ bearing mice are well referenced in the literature and our survival data are consistent with previous publications using this model. For these reasons we present survival data as an indicator of therapeutic efficacy (Figure 5A).

We have now included a third model, the CT26 colon carcinoma, in which treatment was administered to mice bearing established subcutaneous tumours. Tumour growth curves and corresponding survival data are shown in Figures 5B–D. In this model, treatment with the tetravalent anti-mCD27 resulted in complete tumour regression in responding mice. In contrast, only 50% of the responding mice treated with the bivalent anti-mCD27 survived the tumour challenge.

Moreover, we performed additional experiments with the CT26 model to understand the cell types involved in the anti-tumour response. Data presented in Supplemental Figure 5 demonstrates that treatment with tetravalent anti-mCD27 induced an increase in both total and activated (4-1BB⁺ and granzyme B⁺) tumour-infiltrating CD8⁺ T cells without impacting total tumoral CD4⁺ or CD4⁺/Foxp3⁺ Tregs. These data support the hypothesis that tetravalent anti-mCD27 exerts its effects through direct co-stimulation of CD8⁺ T cells.

Dynamic measurements of the antigen-specific response are shown for the B16-OVA model in Figure 4C of the revised manuscript. This was carried out by serial blood sampling. Our analysis of the tumour microenvironment (TME) in the CT26 model was carried out at a single time point and this analysis showed clear differences in the activation and cytotoxicity markers (4-1BB and granzyme B) on CD8 T cells. Given that the TME immune phenotyping experiments require mice to be sacrificed and that our objective of deciphering the mechanism of action of the tetravalent anti-CD27 mAb was achieved; the tetravalent anti-CD27 mAb induced a clear increase in 4-1BB and granzyme B expression on CD8 T cells without Treg depletion, we could not justify to our animal welfare and ethics board conducting additional time points and sacrificing more animals. The increase in the 4-1BB, an activation receptor associated with improved metabolic fitness of CD8 T cells, the upregulation of the cytotoxic molecule granzyme B on CD8 T cells and the lack of Treg depletion provides mechanistic insights of how the tetravalent anti-CD27 exerts its potent anti-tumour effects. These new data are presented in Supplemental Figure 5 of the revised manuscript. Thus, our data provides a mechanistic explanation for the effects independent of whether there are additional

effects on exhaustion markers. It is noteworthy that the relevance of using exhaustion markers in mouse models is questionable and confounded by the fact that T cell exhaustion is limited by the short duration time of tumour growth (days) compared to human tumours (months/years).

4) The toxicity of the tetravalent anti-CD27 antibody has not been assessed. Will the tetravalent antibody have a reduced half-life? The safety and PK data will be critical to the translation of this tetravalent agonist antibody into clinical drug development.

We thank the reviewer for raising this important point. Accordingly, we have determined the serum concentration of the bivalent and tetravalent antibody and present the new data in Supplemental Figure 2. Fitting the data by nonlinear regression analysis provides $t_{1/2}$ values of 2.1 and 1.8 days for the bivalent and tetravalent antibodies, respectively. Furthermore, treatment of CT26-bearing mice with four doses of the tetravalent antibody did not result in a reduction in body weight, which is commonly used as an indicator of toxicity of agonist immunostimulatory mAbs eg. anti-CD40 and anti-4-1BB (new Supplemental Figure 4).

5) There are several clinical candidates, such as Varlilumab, all of which have shown limited efficacy in both preclinical models and patients. To ensure that this study can potentially be translated into clinical drug development, the author should evaluate the efficacy and toxicity of the tetravalent version of Varlilumab in a humanised CD27 mouse model. Several companies offer humanised CD27 mice.

We appreciate the reviewer's constructive comments regarding Varlilumab, a first-generation anti-human CD27 monoclonal antibody that was withdrawn following a lack of efficacy in Phase I/II clinical trials.

In our study, the tetravalent anti-human CD27 antibody was evaluated using the well-characterised hCD27.15 clone. Unlike Varlilumab, which exhibits weak agonist activity, hCD27.15 has been shown to co-stimulate T cell proliferation in vitro in its soluble form (Heckel et al., *Communications Biology*, 2022). This functional difference is likely due to its distinct epitope, which differs from that targeted by Varlilumab. Notably, the same epitope has been used to select other potent anti-CD27 agonists, such as MK-5890 (Guelen et al., *Journal for ImmunoTherapy of Cancer*, 2022), suggesting that binding to this specific region of CD27 imparts agonism.

Furthermore, hCD27.15 exhibits lower affinity and faster dissociation kinetics compared to Varlilumab (Heckel et al., *Communications Biology*, 2022), properties that have recently been linked to enhanced receptor clustering and agonistic activity in other anti-TNFRSF antibodies (Yu et al., *Nature*, 2023). Interestingly, the affinity of hCD27.15 is similar to that of the anti-mouse CD27 antibody used in our study.

In summary, both the epitope specificity and binding kinetics of anti-TNFRSF antibodies are critical determinants of their agonistic function. Our selection of hCD27.15 for engineering into the new tetravalent format was therefore driven by a deliberate effort to incorporate these favourable features. We have revised the manuscript text to better highlight these key features and the selection of hCD27.15 as our mAb of choice.

The use of a humanised CD27 mouse model to evaluate the bivalent and tetravalent anti-human CD27 hIgG1 V11 formats is not straightforward because the V11 Fc mutations, which are designed to enhance binding to human FcγRIIB and reduce binding to the activating human FcγRs, do not exert the same effects on mouse FcγRs.

To illustrate this, we provide the reviewer with K_D values determined by SPR analysis. For binding to **mouse FcγRII**, the K_D values are as follows:

- hIgG1: 2.2×10^{-6} M
- hIgG1 V11: 1.3×10^{-6} M
- mIgG1: 3.9×10^{-7} M

In contrast, for **human FcγRIIB**, hIgG1 and hIgG1 V11 show K_D values of 9.4×10^{-7} M and 7.8×10^{-8} M, respectively. These data highlight that a humanised CD27 mouse model cannot faithfully replicate the physiological interaction between the tetravalent anti-CD27 V11 antibody and human FcγRIIB.

Therefore, to test in a humanised mouse model one would need to generate mice with humanised CD27 and human FcγRs. Generating these double transgenic mice would take between 18-24 months providing they can be sourced from suppliers and collaborators. Moreover, the human FcγR transgenic mice have variable levels of FcγRs expression that do not always reflect physiological levels seen on human leukocytes and human IgG1 has a much shorter half-life in mice.

Given these caveats, we felt it was more appropriate to investigate anti-human CD27 antibodies using our human PBMC assay. This assay employs whole human PBMCs, thereby preserving not only physiological levels of human CD27 expression on T cells but also the native expression profile of human FcγRs on B cells, myeloid cells, and NK cells such as would be present in human blood. This system proved invaluable for distinguishing the functional profiles of the various anti-human CD27 antibody formats, including tetravalent vs. bivalent, and hIgG1 vs. Fc-silent (hIgG1 NA) vs. hIgG1 V11, offering key mechanistic insights that would be obscured in murine models lacking human FcγR biology.

Minor points:

1) Table 1, which model is used to fit the SPR data, 1:2 or 1:1 kinetic model? Is the binding affinity apparent K_D ?

As detailed in the table legend, the bivalent (2:1) fit was used to model affinity. (2:1) has now been included for clarity. Affinity is assumed to be apparent affinity due to multivalent binding, with table 1 and supplementary table 2 (previously supplementary table 1) updated to reflect this.

2)The authors should use a consistent name throughout the manuscript, for example, CD8+ T cells and CD8 T cells.

Noted and changed for consistency.

3) Figure 4C, the tumor volume at the last time point was less than 200mm³. The tumor growth curve should be plotted until some tumors reach 1500 or 2000mm³.

To clarify, tumour size is determined by cross-sectional area and is the product of measurable perpendicular tumour dimensions (length × width). The units on the y-axis of the tumour growth graphs are mm². A cross-sectional area of 200 mm² is approximately equivalent to a volume of 1500 mm³.

4) In lines 271-275 of the Methods section, the author should clarify how many doses of antibody were administered. It would be better to inject OVA with an adjuvant to better induce T cell immunity.

We have added the frequency of antibody dosing to the Methods section. In our study, the CD27 agonist was used as an adjuvant. CD27 agonists exhibit potent adjuvant-like properties for enhancing CD8⁺ T cell responses, surpassing those of conventional adjuvants such as TLR4 agonists (Rowley and Al-Shamkhani, J Immunol, 2004).

6) In the Figure 5 section, the author should discuss that some human CD27 antibodies are less dependent on FcγRIIb cross-linking, but that FcγRIIb cross-linking can further enhance potency, suggesting that the agonistic activity of CD27 antibodies may be influenced by their epitopes. The bivalent antibody and the tetravalent antibody with the Fc V11 variant did not show a significant difference in Figure 5A. What is the reason?

We agree with the reviewer that some antibodies can elicit agonism in their soluble form and are less dependent on FcγR crosslinking, and that this ability is influenced by the specific epitope recognized by the antibody. The anti-human CD27 clone hCD27.15 used in our study is known to target an epitope associated with stronger agonistic activity (please refer to our response to Reviewer 3, Major Point 5, for a detailed discussion on the roles of epitope specificity and affinity). We have updated the manuscript text accordingly in light of the reviewer's comment.

The reviewer also asked, "The bivalent antibody and the tetravalent antibody with the Fc V11 variant did not show a significant difference in Figure 5A. What is the reason?" The tetravalent anti-hCD27 hIgG1 V11 clearly demonstrated greater activity at lower concentrations compared to the bivalent anti-hCD27 hIgG1 V11, as indicated by the highly statistically significant difference shown in Figure 5A (lower right panel). Specifically, the EC₅₀ values for the tetravalent and bivalent V11 antibodies in the presence of FcγRIIB were 0.0002 and 0.008 nM, respectively, a 40-fold difference.

7) In Figure 6C, the video should be provided in supplemental information.

As detailed in the Figure and Methods section, this experiment was conducted as a time course rather than a live video recording, with cells fixed at the indicated time points and subsequently imaged. Representative images are shown in the Figure.

9) For Figure 6E, it would be informative to analyze whether the percentage of GFP⁺ cells shows significant changes after treatment with either bivalent or tetravalent antibodies. Additionally, to enhance data interpretation, both MFI and the percentage of positive cells could be calculated and presented on the y-axis.

We thank the reviewer for their constructive suggestion, and we have now included these graphs as Figure 6F.

10) The Discussion section would benefit from a comprehensive analysis of the potential limitations associated with the tetravalent antibody approach, including but not limited to its structural stability, pharmacokinetic properties, and potential immunogenicity.

We thank the reviewer for this important comment. We have now added new data on the half-life and thermostability of the tetravalent antibody, which show only a modest reduction compared to conventional IgG. These results are included and discussed in the revised manuscript.

Regarding immunogenicity, the tetravalent antibody sequence is composed almost entirely of native human IgG components. As such, the likelihood of eliciting a strong anti-drug immune response is expected to be low. It is worth noting that clinically approved chimeric antibodies such as infliximab and cetuximab, despite containing a significantly higher proportion of non-human (murine) sequence, have been successfully used in patients, further supporting the tolerability of antibody formats with structural modifications. In response, we have added a final paragraph to the

Discussion to address potential limitations associated with the tetravalent antibody, including increased immunogenicity.

11) The term "Finaly" in line 175 requires correction to "Finally." Additionally, we suggest performing a comprehensive spell-check of the entire manuscript to identify and rectify any similar typo.

Noted and corrected.

12) Although Figure S2 shows no aggregation of the tetravalent antibody, this does not mean that it is stable. The thermostability of the tetravalent antibody should be determined.

Thermostability was assessed using nano differential scanning fluorimetry (nanoDSF), which demonstrated that the tetravalent antibody has a melting temperature of 56.7 °C. Although this is slightly lower than that of the IgG molecule (63.5 °C), it is well above physiological temperatures and the standard accelerated storage temperature of 40°C for monoclonal antibodies. These results are summarised in Supplemental Table 1 and discussed in the revised manuscript in the Results and Discussion.

Reviewer #4 (Remarks to the Author):

We would like to thank the reviewer for their constructive comments and suggestions.